# Component-Wise Composite Likelihood Distillation for Censored Time-to-Event Data

Feiyang Deng [1]  Lingfeng Luo [1]  Jiayu Zhou [2]  Kevin He [1]

## Abstract

Accurate survival modeling in biomedical studies is often hindered by rare events, limited effective sample sizes, and settings with limited or partially observed information (e.g., covariates of interest that are difficult or expensive to collect, highly structured sampling designs, or nuisance parameters omitted by conditioning). Knowledge distillation can leverage external predictive information without sharing individual-level data, but existing approaches are largely built for fully specified likelihoods or probability-based survival models and do not extend to settings where outcome distributions are only partially specified. To address this challenge, we propose a knowledge distillation framework based on a composite-likelihood Kullback–Leibler divergence that aligns teacher and student models within components. Our key insight is that, although composite likelihoods do not define a global outcome distribution, each likelihood component induces a well-defined probability model on its restricted outcome space, enabling a principled KL divergence. Simulation studies and biomedical case studies show improved discrimination and predictive accuracy in rare-event, heterogeneous settings without requiring access to external individual-level data.

## 1. Introduction

External predictive information from registries, multi-center collaborations, or historical cohorts is often available and can improve time-to-event modeling when internal data are limited (Jiang et al., 2016; Chen et al., 2021; Sheng et al., 2021; Cheng et al., 2023). A central difficulty is that many

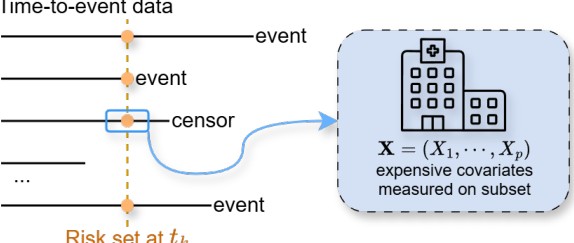

*Figure 1.* Cohort follow-up with time-to-event outcomes and cost-driven selective measurement. While the full cohort is followed over time, expensive covariates are collected only for a subset of individuals around observed failures (e.g., the case and a small number of sampled controls), inducing a nested case–control dataset for downstream analysis.

survival and case–control analyses, including modern deep survival pipelines, are built on locally normalized learning objectives rather than globally normalized outcome models. These objectives arise from both computation and inference. Computationally, events are rare and registries induce extremely large risk sets, so full risk-set normalization is often infeasible and naive mini-batch optimization yields weak signal (Tarkhan & Simon, 2020; Zeng et al., 2026). Inferentially, especially in settings with limited or partially observed information (e.g., covariates that are difficult or expensive to collect, highly structured sampling designs, or nuisance parameters omitted by conditioning), valid analysis frequently conditions away nuisance structure such as stratum-specific intercepts, baseline hazards, censoring, and sampling mechanisms (Breslow, 1974; Cox, 1975; Efron, 1977). Consequently, the identifiable signal is expressed through within-set relative-risk comparisons on comparability sets, and globally normalized per-individual outcome probabilities are not identified. This setting makes external information attractive but complicates transfer, because standard transfer learning and knowledge distillation are typically formulated in terms of globally normalized probabilities or model-aligned representations (Hinton et al., 2015; Gou et al., 2021).

Nested case–control (NCC) sampling (Goldstein & Langholz, 1992) provides a concrete and widely used in-

[1]Department of Biostatistics, University of Michigan, Ann Arbor, MI, USA [2]School of Information, University of Michigan, Ann Arbor, MI, USA. Correspondence to: Kevin He <kevinhe@umich.edu>.

*Proceedings of the $43^{rd}$ International Conference on Machine Learning*, Seoul, South Korea. PMLR 306, 2026. Copyright 2026 by the author(s).

stance of this regime and motivates our development. In many cohort studies, collecting the full set of covariates for every participant at every follow-up time is infeasible because measurements can be expensive, invasive, or available only when individuals return for clinical care; moreover, event rates are often low, so measuring all at-risk individuals yields little additional information relative to the incurred cost (Goldstein & Langholz, 1992; Ernster, 1994; Langholz & Clayton, 1994). A common response is selective measurement, where detailed covariates are collected for failures and for a small subset of controls sampled from the risk set at each failure time (Figure 1). NCC designs formalize this practice by constructing event-centered comparability sets that preserve valid relative-risk information while substantially reducing data collection burden. Modern deep Cox training also exhibits an analogous structure: many implementations use risk-set subsampling and batch-restricted normalization to define event-centered contrastive objectives with sampled denominators that make stochastic optimization feasible in large registries (Katzman et al., 2018; Kvamme et al., 2019). These objectives are typically distinct from the full partial-likelihood loss, but are designed to target the same underlying relative-risk structure. These connections indicate that locally normalized, comparability-set-based objectives are not an edge case but a common operating regime in both classical study designs and scalable deep survival training.

Standard knowledge distillation does not directly apply in this locally normalized regime. Response-based distillation assumes that soft targets are well defined as globally normalized outcome probabilities (Zhao et al., 2022; Wang et al., 2025b), while feature- and relation-based distillation assumes that latent representations or scoring scales are comparable across models (Park et al., 2019; Yang et al., 2023). Under conditional, partial, and composite likelihood objectives, probabilities are normalized only within each likelihood component, and the only stable and identifiable predictive structure is relative risk within a comparability set (Lindsay, 1988; Varin et al., 2011; Clayton & Hills, 2013). For example, in the Cox partial likelihood, risk-set–constant nuisance terms (e.g., the baseline hazard) cancel in the within-set normalization, so the objective identifies only within-set log-risk differences (Kalbfleisch & Prentice, 2002). Consequently, a teacher's globally calibrated event probability (or absolute risk) is generally not identifiable under the student's objective. Accordingly, distillation targets defined on globally calibrated probabilities are mismatched to locally normalized objectives, and naive cross-model alignment (e.g., representation matching) may transfer non-identifiable nuisance variation, leading to brittle behavior under heterogeneity.

Our key observation resolves this mismatch. Although conditional, partial, and composite likelihoods do not define

a globally normalized predictive distribution, each likelihood component induces a normalized distribution on its restricted outcome space. This induced component distribution provides the appropriate object for model alignment. We therefore propose a component-wise distillation: for each likelihood component, we construct the teacher-induced and student-induced component distributions and minimize a valid Kullback–Leibler divergence between them. The resulting objective transfers an external relative-risk structure, which is the information identified under locally normalized inference, without requiring globally normalized probabilities, shared parameterizations, or aligned latent representations. Because components correspond to event-centered comparability sets, the method is compatible with stochastic optimization and inherits the scalability benefits of NCC-style sampling and mini-batch risk-set approximations.

In summary, this work makes three contributions. First, we provide a unifying probabilistic view of locally normalized learning over comparability sets, encompassing objectives from classical survival and case–control analysis (e.g., Cox partial likelihood and matched designs), study-design–induced sampling (e.g., nested case–control), and scalable deep-survival training via risk-set subsampling. Second, we introduce a general component-wise Kullback–Leibler distillation framework for this class of objectives that remains compatible with flexible deep models trained with stochastic optimization. Third, through simulations and biomedical case studies, we demonstrate that component-wise distillation improves discrimination and predictive accuracy in data-limited, heterogeneous, and rare-event settings while requiring no access to external individual-level data.

**Code Availability.** An accompanying code repository is available at `https://github.com/yatoka233/CompositeDistill`.

## 2. Related Work

**Distillation for Ranking and Retrieval.** Knowledge distillation has been studied for ranking, retrieval, and recommendation by aligning teacher and student behaviors on a query-specific candidate list, for example, through softened relevance scores, pairwise preferences, or listwise ranking distributions (Tang & Wang, 2018; Lee et al., 2019; Reddi et al., 2021). Although these methods operate on sets, the data structure and inferential target differ from ours. In ranking, the candidate list is part of the problem definition for a query, and the target is an ordering or relevance pattern on that list. In our setting, comparability sets are induced by study design and censoring-aware risk-set construction (e.g., matched strata, risk sets, sampled risk sets), and within-set normalization is required to condition away nuisance structure. The target is therefore local relative-risk information

within each comparability set rather than a globally defined ranking distribution. Thus, our distillation is formulated on the locally normalized likelihood components implied by the design rather than on query-level ranking distributions.

**Transfer Learning in Survival Analysis.** Although transfer learning methods (Chen et al., 2021; Sheng et al., 2021; Cheng et al., 2023; Wang et al., 2025a) have been proposed for survival data integration, these techniques are largely developed around specific survival regression models. Several works incorporate external prognostic information into time-to-event modeling using KL-style alignment. Most closely related, Wang et al. (2026) proposed CoxKL, which regularizes a Cox model via a KL divergence defined on Cox partial-likelihood objects. These results demonstrate the promise of KL alignment under heterogeneity, but their formulations are tied to classical modeling objects, such as Cox models or discrete-time models.

Our framework targets locally normalized objectives induced by study design rather than a specific globally normalized model, with a focus on practical regimes where key information is constrained by design or availability (e.g., expensive covariates, structured sampling, or nuisance features removed by conditioning). We place these objectives within a unified component-level framework, where the learning problem is decomposed into likelihood components defined in comparability sets. We therefore define distillation at the component level by aligning teacher- and student-induced targets within each set via a component-wise KL divergence. This yields a unified design-aware transfer principle that extends beyond Cox partial likelihood.

## 3. Preliminary: Composite Likelihood

Let $\{O_i\}_{i=1}^n$ denote the observed data, where $O_i$ collects all recorded information for unit $i$ (e.g., covariates, outcomes, censoring indicators, and design variables). We consider a working full-data model with density (or mass function) $f(O \mid \boldsymbol{\beta}, \boldsymbol{\alpha})$, where $\boldsymbol{\beta}$ denotes the parameter of scientific interest and $\boldsymbol{\alpha}$ collects nuisance components (e.g., baseline terms, stratum effects, or sampling mechanisms). In many stratified, semiparametric, or sampling-based designs, directly maximizing the full likelihood is impractical because $\boldsymbol{\alpha}$ is high-dimensional or the likelihood involves intractable normalizing constants.

Composite likelihood replaces the full likelihood with a sum of tractable, locally defined components (Lindsay, 1988; Varin & Vidoni, 2005; Varin et al., 2011). Specifically, the design induces $K$ local comparison problems indexed by $k = 1, \ldots, K$. For each component $k$, let $\mathcal{I}^{(k)} \subset \{1, \ldots, n\}$ be the comparability set of units compared within that component (e.g., a matched stratum or a risk set). Let $\mathbf{X}_i$ denote the covariates extracted from

$O_i$, and write $\mathbf{X}^{(k)} = \{\mathbf{X}_i : i \in \mathcal{I}^{(k)}\}$ for the covariates restricted to the set $\mathcal{I}^{(k)}$. The observed event configuration on $\mathcal{I}^{(k)}$ is encoded by a binary label vector $\mathbf{Y}^{(k)} = (Y_i^{(k)})_{i \in \mathcal{I}^{(k)}}$, where $Y_i^{(k)} \in \{0, 1\}$ indicates whether unit $i$ is treated as an event (case) within component $k$. The design further specifies a conditioning event that restricts the admissible label patterns to a set $\mathcal{Y}^{(k)} \subseteq \{0, 1\}^{|\mathcal{I}^{(k)}|}$. A common instance is a fixed case-count constraint,

$$\mathcal{Y}^{(k)}(t_k) = \left\{ \mathbf{y} \in \{0, 1\}^{|\mathcal{I}^{(k)}|} : \sum_{i \in \mathcal{I}^{(k)}} y_i = t_k \right\},$$

which enumerates all allocations of $t_k$ events within $\mathcal{I}^{(k)}$. The restricted outcome space is component-specific: it is defined after conditioning on the design information for component $k$, such as the number of cases in a matched stratum or the number of failures at a Cox event time. It should therefore not be interpreted as a globally normalized outcome space for the full cohort.

We focus on conditional-type composite likelihoods in which each component defines a normalized probability model on $\mathcal{Y}^{(k)}$.

## 4. Method

### 4.1. Locally Normalized Component Models

Across conditional logistic regression (CLR), Cox partial likelihood, and nested case–control designs, inference is driven by within-set contrasts: within each comparability set, only relative differences in risk scores are identifiable, while terms shared by all units in the set are removed by conditioning. We formalize this invariance by assuming that, for each component $k$ and each unit $i \in \mathcal{I}^{(k)}$, the model assigns an unnormalized log-score of the additive form

$$u_k(i; \boldsymbol{\beta}) = b_k + s_{\boldsymbol{\beta}}(\mathbf{X}_i). \tag{1}$$

Here $\mathbf{X}_i \in \mathbb{R}^p$ denotes the covariate vector for unit $i$, $s_{\boldsymbol{\beta}}(\cdot)$ is a learnable real-valued scoring function parameterized by $\boldsymbol{\beta}$ (e.g., linear, generalized additive, or a neural network), and $b_k$ is a component-specific term common to all units in component $k$. The interpretation of $b_k$ depends on the design: in CLR it corresponds to a stratum intercept; in Cox-type components it corresponds to the log baseline hazard at the event time; and in NCC it may also absorb set-level factors induced by sampling or matching, provided they are common to all members of the sampled set. Because such terms are shared within the component, they are not identifiable from within-set comparisons.

Given the restricted outcome space $\mathcal{Y}^{(k)}$ defined by the design (Section 3), the component likelihood is defined as the conditional distribution on $\mathcal{Y}^{(k)}$ induced by the unnormalized scores in (1). For the conditional-type components considered below, the conditioning event fixes the

component event count: there exists an integer $m_k$ such that $\sum_{i \in \mathcal{I}^{(k)}} y_i = m_k$ for all $\mathbf{y} \in \mathcal{Y}^{(k)}$. This fixed-count condition removes component-shared offsets such as $b_k$ under local normalization. Specifically, for a label vector $\mathbf{y}^{(k)} = (y_i^{(k)})_{i \in \mathcal{I}^{(k)}} \in \mathcal{Y}^{(k)}$, we define

$$f_k(\mathbf{y}^{(k)}; \boldsymbol{\beta}) = \frac{\exp\left\{ \sum_{i \in \mathcal{I}^{(k)}} y_i^{(k)} s_{\boldsymbol{\beta}}(\mathbf{X}_i) \right\}}{Z_k(\boldsymbol{\beta})}, \qquad (2)$$

$$Z_k(\boldsymbol{\beta}) = \sum_{\mathbf{u} \in \mathcal{Y}^{(k)}} \exp\left\{ \sum_{i \in \mathcal{I}^{(k)}} u_i\, s_{\boldsymbol{\beta}}(\mathbf{X}_i) \right\}.$$

The shared term $b_k$ does not appear in (2) because, under the fixed-count condition, the factor $\exp(b_k m_k)$ is common to the numerator and every term in the denominator and therefore cancels. Thus $f_k(\cdot; \boldsymbol{\beta})$ depends only on within-set contrasts encoded by $s_{\boldsymbol{\beta}}(\cdot)$.

Stacking the component log-likelihoods yields the composite objective

$$\ell(\boldsymbol{\beta}) = \sum_{k=1}^{K} \log f_k(\mathbf{y}^{(k)}; \boldsymbol{\beta}). \qquad (3)$$

In this section, the role of composite likelihood is purely foundational: it formalizes training under designs where the global nuisance structure is not specified or estimable, while preserving the same within-set normalization that underlies CLR, Cox, and NCC.

### 4.2. Component-Wise Knowledge Distillation

Our goal is to transfer predictive structure from a teacher to a student when the student is trained under within-set normalization. In standard i.i.d. distillation, the KL divergence is defined between per-instance predictive distributions. In conditional, partial, and pseudo-likelihood settings, however, the student is not trained on globally normalized per-instance probabilities; it is identified only through the locally normalized component distributions induced on the restricted spaces $\{\mathcal{Y}^{(k)}\}_{k=1}^{K}$. We therefore distill at the same level at which the student is identified, namely the component distributions $\{f_k(\cdot; \boldsymbol{\beta})\}$ defined in (2).

**Teacher-Induced Component Distributions.** Assume the teacher assigns each unit $i$ a real-valued score $\tilde{s}(\mathbf{X}_i)$. The teacher score may be obtained from a model trained on a different cohort and may live on an arbitrary scale; importantly, we do not require the teacher to specify a coherent global likelihood on the student cohort. Given a comparability set $\mathcal{I}^{(k)}$ and its restricted outcome space $\mathcal{Y}^{(k)}$, we construct a teacher-induced locally normalized distribution on the same space $\mathcal{Y}^{(k)}$ by exponentiating and normalizing

the teacher scores within the set:

$$\tilde{f}_k(\mathbf{y}^{(k)}) = \frac{\exp\left\{ \sum_{i \in \mathcal{I}^{(k)}} y_i^{(k)} \tilde{s}(\mathbf{X}_i) \right\}}{\tilde{Z}_k}, \qquad (4)$$

$$\tilde{Z}_k = \sum_{\mathbf{u} \in \mathcal{Y}^{(k)}} \exp\left\{ \sum_{i \in \mathcal{I}^{(k)}} u_i\, \tilde{s}(\mathbf{X}_i) \right\}.$$

This construction is intentionally local: it does not introduce or rely on any student-side nuisance terms (such as $b_k$ in (1)), and it does not require the teacher to provide globally calibrated event probabilities. Additive shifts of the teacher scores within a component cancel under local normalization, while multiplicative scaling changes the sharpness of the induced component distribution and is treated as part of the teacher signal rather than as a globally calibrated probability scale. The only requirement is set compatibility: for each component $k$, the teacher can evaluate $\tilde{s}(\mathbf{X}_i)$ for all units $i \in \mathcal{I}^{(k)}$, so that both $\tilde{f}_k(\cdot)$ and $f_k(\cdot; \boldsymbol{\beta})$ are probability distributions on the same restricted space $\mathcal{Y}^{(k)}$.

**Composite KL and the Distilled Objective.** For each component $k$, both the student $f_k(\cdot; \boldsymbol{\beta})$ and the teacher $\tilde{f}_k(\cdot)$ define probability distributions on the same restricted space $\mathcal{Y}^{(k)}$. We therefore define a component-wise KL divergence and aggregate it across components:

$$\mathrm{D}(\tilde{f} \| f_{\boldsymbol{\beta}}) = \sum_{k=1}^{K} \mathrm{KL}\left( \tilde{f}_k \,\|\, f_k(\cdot; \boldsymbol{\beta}) \right), \qquad (5)$$

$$\mathrm{KL}\left( \tilde{f}_k \,\|\, f_k(\cdot; \boldsymbol{\beta}) \right) = \sum_{\mathbf{y} \in \mathcal{Y}^{(k)}} \tilde{f}_k(\mathbf{y}) \log \frac{\tilde{f}_k(\mathbf{y})}{f_k(\mathbf{y}; \boldsymbol{\beta})}. \qquad (6)$$

We train the student by maximizing a penalized composite log-likelihood,

$$\ell_\eta(\boldsymbol{\beta}) = \ell(\boldsymbol{\beta}) - \eta\, \mathrm{D}(\tilde{f} \| f_{\boldsymbol{\beta}}), \qquad (7)$$

where $\eta \geq 0$ controls the distillation strength.

**Proposition 4.1** (Equivalent soft-label form). *Suppose that each component likelihood is defined by* (2) *and that the teacher-induced component distribution $\tilde{f}_k$ is defined on the same restricted outcome space $\mathcal{Y}^{(k)}$. Let*

$$\tilde{p}_{ik} = \mathbb{E}_{\tilde{f}_k}\left[ Y_i^{(k)} \right] = \sum_{\mathbf{y}^{(k)} \in \mathcal{Y}^{(k)}} y_i^{(k)} \tilde{f}_k(\mathbf{y}^{(k)})$$

*denote the teacher-implied marginal event probability of unit $i$ within component $k$. Then the penalized objective* (7) *is equivalent, up to additive constants independent of $\boldsymbol{\beta}$, to*

$$\ell_\eta(\boldsymbol{\beta}) = \sum_{k=1}^{K} \left[ \sum_{i \in \mathcal{I}^{(k)}} \left( y_i^{(k)} + \eta\, \tilde{p}_{ik} \right) s_{\boldsymbol{\beta}}(\mathbf{X}_i) \right.$$

$$\left. - (1 + \eta) \log Z_k(\boldsymbol{\beta}) \right] + Const. \qquad (8)$$

*Equivalently, after multiplying the objective by the positive constant $(1 + \eta)^{-1}$, maximizing (8) is equivalent to maximizing the original component objective with soft labels*

$$\tilde{y}_i^{(k)} = \frac{1}{1+\eta} y_i^{(k)} + \frac{\eta}{1+\eta} \tilde{p}_{ik}. \tag{9}$$

Proposition 4.1 shows that component-wise KL distillation has a simple soft-label interpretation: the empirical event indicator within each component is blended with the teacher-implied marginal event probability on the same restricted outcome space. Thus $\eta$ continuously interpolates between empirical training ($\eta = 0$) and full within-set imitation of the teacher ($\eta \to \infty$). The proof is provided in Appendix A.1. Importantly, distillation is defined entirely on the locally normalized outcome spaces $\{\mathcal{Y}^{(k)}\}_{k=1}^K$. Consequently, nuisance terms removed by conditioning in the base objective are not reintroduced by the KL penalty.

Appendix A.2 further provides a local excess-risk expansion showing that the component-wise KL penalty induces a bias–variance tradeoff: a teacher that is sufficiently aligned with the target component structure can reduce estimation variance, whereas teacher–target mismatch contributes bias.

### 4.3. Special Cases: CLR, Cox, and NCC

The general construction is instantiated by specifying, for each component $k$, a comparability set $\mathcal{I}^{(k)}$ and a restricted outcome space $\mathcal{Y}^{(k)}$, together with the within-set scoring decomposition in (1). We summarize three canonical designs below.

**Conditional Logistic Regression (CLR).** Each component $k$ is a matched stratum. The comparability set $\mathcal{I}^{(k)}$ contains the $n_k$ units in stratum $k$, and the restricted space $\mathcal{Y}^{(k)}(t_k)$ fixes the number of cases $t_k$ within the stratum. A convenient working model is the stratified logistic form

$$logit\{\Pr(Y_i^{(k)} = 1 \mid \mathbf{X}_i)\} = \alpha_k + s_{\boldsymbol{\beta}}(\mathbf{X}_i), \quad i \in \mathcal{I}^{(k)},$$

so that the unnormalized score in (1) can be interpreted as the log-odds, with $\alpha_k$ a stratum-specific intercept. Conditioning on the case count $t_k$ removes $\alpha_k$, hence the resulting component likelihood depends only on within-stratum contrasts of $s_{\boldsymbol{\beta}}(\mathbf{X}_i)$.

**Cox Partial Likelihood.** We now consider right-censored survival data with latent event times $T_i^*$, censoring times $C_i$, observed times $T_i = \min(T_i^*, C_i)$, and event indicators $\Delta_i = \mathbb{1}(T_i^* \le C_i)$. Let $\tau_1 < \cdots < \tau_K$ denote the distinct observed event times. Each component $k$ corresponds to the risk set at time $\tau_k$,

$$\mathcal{I}^{(k)} = R(\tau_k) = \{i : T_i \ge \tau_k\},$$

and the restricted space $\mathcal{Y}^{(k)}(d_k)$ fixes the number of failures $d_k$ at $\tau_k$ (in particular, $d_k = 1$ under no ties). Under the Cox proportional hazards model,

$$\lambda_i(t) = \lambda_0(t) \exp\{s_{\boldsymbol{\beta}}(\mathbf{X}_i)\},$$

the log hazard at $\tau_k$ decomposes as

$$\log \lambda_i(\tau_k) = \log \lambda_0(\tau_k) + s_{\boldsymbol{\beta}}(\mathbf{X}_i),$$

so that the unnormalized within-set score in (1) can be taken as $u_k(i; \boldsymbol{\beta}) = \log \lambda_0(\tau_k) + s_{\boldsymbol{\beta}}(\mathbf{X}_i)$. Conditioning on the event configuration within $R(\tau_k)$ eliminates the baseline term, yielding the partial likelihood component that depends only on within-risk-set contrasts of $s_{\boldsymbol{\beta}}(\mathbf{X}_i)$.

**NCC and Matched NCC.** Each component corresponds to a sampled (and possibly matched) risk set constructed around an observed failure time $\tau_k$. Let $i_k$ denote the failing case, and let $\mathcal{C}_k$ be $n_c$ controls sampled from an eligible pool $R_{\mathrm{elig}}(\tau_k) \subseteq R(\tau_k)$, yielding the comparability set $\mathcal{I}^{(k)} = \{i_k\} \cup \mathcal{C}_k$. We view this as a risk-set approximation of the Cox model: within $\mathcal{I}^{(k)}$, the relative failure propensity is still governed by proportional hazards, so the local unnormalized score takes the additive form

$$u_k(i; \boldsymbol{\beta}) = b_k + s_{\boldsymbol{\beta}}(\mathbf{X}_i), \quad b_k = \log \lambda_0(\tau_k),$$

and the component likelihood depends only on within-set contrasts of $s_{\boldsymbol{\beta}}(\mathbf{X}_i)$ because $b_k$ cancels under the normalization in (2). Matched NCC is obtained by defining $R_{\mathrm{elig}}(\tau_k)$ through matching constraints (e.g., donor strata, center, or calendar time), which may introduce additional shared baseline terms within the matched eligibility class; these are absorbed into $b_k$ and likewise cancel within each $\mathcal{I}^{(k)}$. Therefore, once the sampling/matching step induces $\mathcal{I}^{(k)}$ (and hence $\mathcal{Y}^{(k)}$), both the locally normalized component model and our distillation objective apply without modification.

## 5. Experiments

We combine controlled simulations and a real transplant-registry case study for complementary purposes. Simulations isolate the key mechanisms motivating our method, including rare events and limited internal cohorts, heterogeneous teachers with incomplete or shifted covariates, and uncertainty in the scale or calibration of external scores, while keeping the data-generating process fixed. The Organ Procurement and Transplantation Network (OPTN) study evaluates practical utility in two complementary real-world settings. First, we study transfer from the large Kidney–alone (KI) cohort to the much smaller and systematically different Kidney–pancreas (KP) cohort, reflecting a common deployment scenario where the target population is data-limited and heterogeneous. Second, we study a deployable recipient-only risk score for KP using a donor-matched

NCC design to enforce comparability and mitigate donor-driven selection and outcome mechanisms.

## 5.1. Metrics

We evaluate right-censored survival predictions using censoring-aware metrics that assess both discrimination and calibration. For discrimination, we report Harrell's concordance index ($C$-index) (Harrell et al., 1982), a stratified $C$-index that restricts comparisons to units within the same comparability set, and the mean time-dependent AUC (MeanAUC) (Uno et al., 2007; Hung & Chiang, 2010; Lambert & Chevret, 2016). For calibration and overall predictive accuracy, we use the integrated Brier score (IBS) (Graf et al., 1999; Gerds & Schumacher, 2006; Kvamme et al., 2019). We also report predictive deviance, defined as the negative log-likelihood on the held-out test set, as a likelihood-based measure of goodness of fit (Burnham & Anderson, 2002; Tutz et al., 2016). Formal definitions and implementation details are provided in Appendix B.

## 5.2. Tuning Hyperparameters

All hyperparameters are selected using five-fold cross-validation on the corresponding non-test data. Automated search is performed with `Optuna` using the TPE sampler and a fixed budget of 20 trials, so that the tuning budget is identical across methods (Akiba et al., 2019). Teacher and student models are tuned separately within their respective cohorts. We consider a common search space covering network depth, hidden width, dropout rate, batch size, and learning rate, and additionally $\eta$ for distilled models. Unless stated otherwise, we use a two-stage procedure for distilled students: we first tune architecture and optimization hyper-parameters, and then, holding them fixed, tune $\eta$ by five-fold cross-validation. Additional implementation and tuning settings are reported in Appendix C.1.

## 5.3. Simulation

We evaluate component-wise distillation through controlled simulations that mirror learning from sampled comparability sets, as in nested case–control training. The goal is to quantify when a teacher can improve a data-limited student under within-set normalization, and to stress-test robustness to two practical mismatches: (i) cohort heterogeneity between teacher and student, and (ii) model heterogeneity, including differences in feature availability and in the scale or calibration of the teacher risk score.

**Simulated NCC Datasets.** We generate a large teacher cohort ($n = 10{,}000$), a smaller student cohort ($n = 500$), and an independent test cohort ($n = 5{,}000$) from the same underlying distribution. To construct NCC training data, we apply risk-set sampling to the teacher and student co-

horts. At each observed failure time $t_i$ for case $i$, we sample $n_c = 16$ controls without replacement from the corresponding risk set $R(t_i) \setminus \{i\}$, forming a comparability set with exactly one event and $n_c$ non-events. The resulting collection of sampled sets defines the composite-likelihood components used to fit the teacher and student under the NCC objective, and it also defines the comparability sets used by component-wise distillation. The test cohort is not subjected to NCC sampling; we evaluate all methods on the full test cohort to reflect population-level predictive performance. Complete data-generation and evaluation details are provided in Appendix C.2.1.

**Distillation Under Varying Teacher Covariate Quality.** To quantify how component-wise distillation depends on the teacher's informativeness, we vary the covariates available to the teacher while keeping the student feature set fixed. We consider three teacher settings: (i) good quality, trained with all covariates $\{X_1, \ldots, X_{12}\}$; (ii) fair quality, trained with $\{X_1, \ldots, X_9\}$; and (iii) poor quality, trained with $\{X_1, \ldots, X_6\}$.

Figure 2 shows a clear monotone pattern. With a good-quality teacher, component-wise distillation yields substantial gains in both discrimination (C-index and MeanAUC) and likelihood-based fit (predictive deviance), moving the student toward the teacher reference. As teacher quality drops from good to fair to poor, these gains attenuate, consistent with weaker teacher signal. Importantly, across settings the distilled student remains competitive with, and typically improves upon, the internal-only student, indicating that the within-set objective can still extract useful guidance even from an imperfect teacher. Sensitivity to the distillation weight $\eta$ was also examined in Appendix D.1.1.

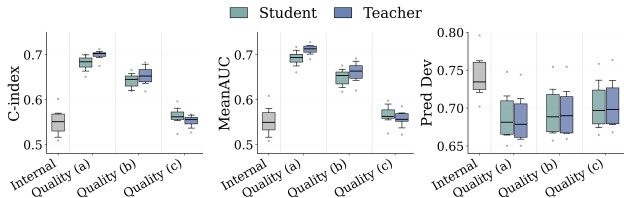

*Figure 2.* Test-set C-index, mean time-dependent AUC, and predictive deviance under different teacher covariate quality settings. Results were summarized over 20 simulation replicates.

**Robustness to Teacher Score Transformations.** A key appeal of component-wise distillation under locally normalized objectives is that learning is driven by within-set contrasts rather than the absolute scale of the teacher score. In practice, teacher outputs can vary across implementations and populations, and may only be meaningful up to simple reparameterizations such as rescaling or shifting, or even more general monotone transforms. To probe sensitivity to these discrepancies, we apply deterministic transformations

*Table 1.* Sensitivity of component-wise distillation to teacher score transformations on the simulated test set. We report mean (SD) over 20 replicates for the internal-only baseline and students distilled from the same teacher after applying different transformations to its risk score (affine scaling, affine shift, and a nonlinear monotone transform).

| METRICS | INTERNAL | SCALE ($\times 0.5$) | EXPONENTIAL | SHIFT ($+2$) | ORIGINAL |
|---|---|---|---|---|---|
| C-INDEX $\uparrow$ | 0.550 (0.027) | 0.677 (0.016) | 0.635 (0.016) | 0.678 (0.018) | 0.681 (0.015) |
| MEANAUC $\uparrow$ | 0.552 (0.030) | 0.687 (0.017) | 0.640 (0.015) | 0.688 (0.019) | 0.692 (0.016) |
| IBS $\downarrow$ | 0.072 (0.003) | 0.064 (0.002) | 0.065 (0.002) | 0.064 (0.002) | 0.064 (0.003) |
| PRED DEV $\downarrow$ | 0.742 (0.026) | 0.684 (0.023) | 0.690 (0.022) | 0.683 (0.023) | 0.688 (0.029) |

to the teacher risk score before distillation, including affine scaling, affine shifting, and an exponential monotone transformation that preserves ranking but alters within-set score gaps, and then refit the student.

Table 1 shows that performance is largely stable under affine transformations. Across scaling and shifting, C-index, MeanAUC, IBS, and predictive deviance remain comparable to using the original teacher score, with only minor fluctuations. The nonlinear monotone transform induces modest degradation, but the distilled student still substantially outperforms the internal-only baseline. Overall, these results suggest that component-wise distillation is not tightly tied to the teacher's absolute score scale and primarily exploits the relative ordering and separation the teacher induces within each comparability set.

**Distillation from a Teacher with Covariate Shift.** We next evaluate component-wise distillation when the teacher is trained on a source population whose covariate distribution differs from the student cohort, introducing population-level heterogeneity beyond teacher misspecification. We generate two shifted teacher populations (moderate and severe; Appendix C.2.2), fit a teacher on each shifted cohort, and distill to a student trained on the original target cohort using the same NCC construction.

Figure 3 summarizes test-set performance. With a moderately shifted teacher, distillation remains effective: the student improves over internal-only training across discrimination and calibration metrics and approaches the teacher reference. Under severe shift, gains attenuate and become less uniform, consistent with weaker cross-population alignment; however, the student remains competitive with the internal baseline and shows no consistent deterioration. Overall, these results suggest that component-wise distillation tolerates moderate covariate shift, with benefits diminishing as the teacher becomes increasingly out-of-domain.

### 5.4. OPTN Real Data Application

This case study is motivated by two practical challenges in transplant prediction. First, clinically meaningful subpopulations are often too small for stable and well-calibrated time-to-event modeling when trained purely in-domain.

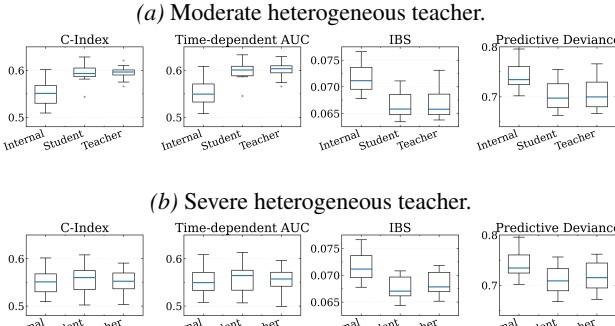

*(a) Moderate heterogeneous teacher.*

*(b) Severe heterogeneous teacher.*

*Figure 3.* Distillation under covariate shift in the NCC simulation. Boxplots summarize test-set performance over 20 replicates when the teacher is trained on a shifted source population and the student is trained on the original target cohort. Panel (a) uses a moderately shifted teacher population and panel (b) uses a severely shifted teacher population.

Kidney–pancreas (KP) recipients are systematically different from kidney-alone (KI) recipients in indications and baseline profiles, yet KP outcomes are rarer and the available KP cohort is much smaller. As a result, flexible survival models that are competitive in large cohorts can become statistically and numerically fragile in KP, motivating a transfer setting in which a teacher trained on the large KI cohort provides guidance to a student trained on KP while allowing for cross-population heterogeneity. Second, deployment constraints often limit which covariates are available at prediction time. In particular, recipient information may be available earlier or more reliably than donor information, but KP outcomes depend on both; naive recipient-only modeling can therefore be confounded by donor-driven selection and outcome mechanisms, motivating a donor-matched NCC design that enforces comparability through donor similarity.

We use de-identified data from the Organ Procurement and Transplantation Network (OPTN) Standard Transplant Analysis and Research (STAR) files for adult recipients transplanted between 2015 and 2019, focusing on one-year post-transplant risk of a composite endpoint (death or graft failure) among KP recipients. We treat KI as the teacher cohort ($n = 45{,}738$, event rate 6.0%) and KP as the student cohort

$(n = 3,141$, event rate $4.1\%$), and evaluate two complementary analyses that reflect distinct deployment goals.

**Transfer for Post-Transplant Outcome Prediction.**
First, we study cross-cohort transfer between Cox-type survival models using donor and recipient covariates in both cohorts. This analysis targets a central bottleneck in KP modeling: limited sample size makes it difficult to reliably learn flexible risk functions from KP outcomes alone. We ask whether component-wise distillation improves KP discrimination and fit relative to internal-only training, while avoiding over-imitation when KI and KP exhibit genuine heterogeneity.

Figure 4 summarizes performance on the KP test cohort for the internal-only KP model, the distilled KP student, and the KI-trained teacher. Component-wise distillation yields consistent gains over internal-only training across all metrics, improving discrimination (higher C-index and MeanAUC) and overall predictive accuracy (lower IBS and predictive deviance). Notably, although the teacher is trained out of domain (KI rather than KP), the distilled student matches or exceeds the teacher on the KP evaluation set: it attains comparable discrimination while achieving the best predictive deviance and IBS. This pattern aligns with the intended behavior of within-set distillation: the teacher provides a strong inductive signal, while the student remains anchored to KP outcomes and can recalibrate to the target cohort rather than inheriting the teacher's risk scale.

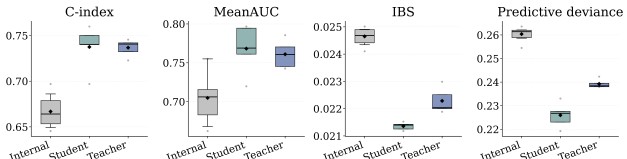

*Figure 4.* OPTN case study (KI → KP): test-set performance on the KP cohort for internal-only training (Internal), the KI-trained teacher (Teacher), and the distilled KP student (Student). We report C-index and MeanAUC (higher is better) together with IBS and predictive deviance (lower is better).

**Recipient-Only Student via Donor-Matched NCC.** Our second analysis targets a deployment-motivated goal: learning a recipient-only KP risk score. In practice, many decisions are made on the recipient side (such as counseling, triage, or candidate ranking), because recipients are evaluated and prioritized in advance to allocate opportunities for high-quality donors. Accordingly, recipient-only models intentionally rely on recipient information, providing a fast and standardized ranking signal that can be computed immediately and applied consistently across workflows.

The key methodological challenge is that KP outcomes are driven by both recipient and donor factors, and donor characteristics also influence which donor–recipient pairs are

realized. As a result, a naive recipient-only Cox model may appear predictive while partially reflecting donor-driven selection and outcome mechanisms. Rather than ignoring donor structure, we enforce comparability through the study design. Specifically, we construct a donor-matched nested case–control objective within KP, where each case is compared only to controls sampled from recipients with similar donor characteristics, stratified by donor age, hypertension, and diabetes. This yields a recipient-only student objective that adjusts for key donor factors through matching while preserving within-set normalization.

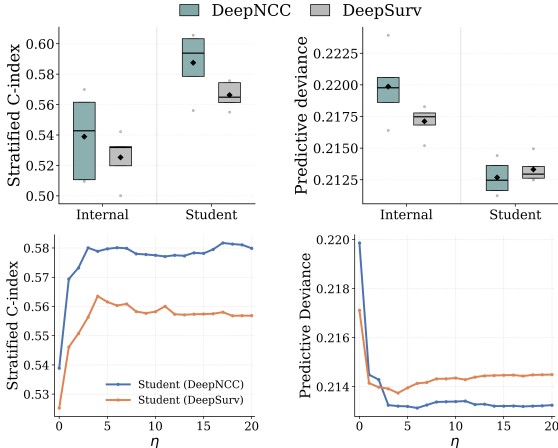

*Figure 5.* Recipient-only KP transfer via donor-matched NCC. **Top:** stratified C-index (left) and stratified predictive deviance (right) for internal-only and distilled students under two deep objectives: DeepSurv trained with the Cox partial-likelihood objective on the full KP cohort, and DeepNCC trained on donor-matched NCC components. **Bottom:** sensitivity of the same metrics to the distillation weight $\eta$ for the two student objectives.

Figure 5 summarizes results by contrasting two deep student objectives with MLP risk-score backbones: DeepSurv, trained with the Cox partial-likelihood objective on the full KP cohort, and DeepNCC, trained on donor-matched NCC components constructed around KP failures. In both cases, using the KI-trained teacher improves performance relative to the corresponding internal-only student. Notably, under recipient-only covariates, the donor-matched DeepNCC student achieves stronger stratified discrimination and lower stratified predictive deviance than the DeepSurv student, consistent with the view that matched within-set comparisons better isolate recipient-driven relative-risk structure instead of conflating it with donor-driven mechanisms. The bottom panels show sensitivity to the distillation weight $\eta$: DeepNCC improves rapidly for small positive $\eta$ and remains stable over a broad range of values, whereas DeepSurv reaches a lower discrimination plateau and less favorable predictive deviance. These results suggest that teacher guidance transfers more directly when the student objective is defined on donor-matched comparability sets.

*Table 2.* KI → KP analysis: comparison of transfer mechanisms in the donor-matched recipient-only setting. All metrics are computed under donor-matched evaluation: stratified C-index and stratified predictive deviance. The upper block reports Cox-compatible baselines for a linear Cox student, while the lower block reports flexible MLP-based DeepSurv and DeepNCC students. CoxKL is reported only for the Cox-compatible student because it is tied to Cox partial-likelihood objects.

| Transfer method | Cox-type objective | | NCC-type objective | |
|---|---|---|---|---|
| | Strat. C-index ↑ | Strat. deviance ↓ | Strat. C-index ↑ | Strat. deviance ↓ |
| **Cox-compatible baselines (linear Cox student)** | | | | |
| Internal-only (Cox) | 0.5382 (0.0113) | 0.2156 (0.0017) | N/A (Cox-specific baseline) | N/A (Cox-specific baseline) |
| CoxKL | 0.5467 (0.0229) | **0.2128 (0.0017)** | N/A (Cox-specific baseline) | N/A (Cox-specific baseline) |
| **Flexible student models (MLP backbones)** | | | | |
| Internal-only | 0.5253 (0.0145) | 0.2171 (0.0011) | 0.5389 (0.0251) | 0.2199 (0.0025) |
| Fine-tune | 0.5297 (0.0187) | 0.2159 (0.0026) | 0.5458 (0.0315) | 0.2170 (0.0023) |
| MSE distill | 0.5533 (0.0122) | 0.2133 (0.0011) | 0.5818 (0.0196) | 0.2132 (0.0018) |
| Component-wise KL | **0.5663 (0.0079)** | 0.2133 (0.0009) | **0.5875 (0.0183)** | **0.2127 (0.0012)** |

**Comparison of Transfer Mechanisms.** To further separate the effect of the transfer mechanism from the choice of student objective, Table 2 compares internal-only training, fine-tuning, MSE-based distillation, CoxKL, and the proposed component-wise KL under the same donor-matched evaluation protocol. The transfer baselines are defined as follows. **Fine-tuning** initializes the KP student from the KI teacher and then trains on KP outcomes using the corresponding student objective. Because the KI teacher uses both donor and recipient covariates while the KP student is restricted to recipient covariates, we transfer only the compatible hidden-layer parameters and randomly initialize the recipient-only input layer. **MSE distillation** adds an $\ell_2$ score-matching penalty between the recipient-only student risk score and the KI teacher risk score, combined with the KP objective. The teacher score is used only during training, so the final student remains recipient-only at evaluation.

For Cox-compatible comparisons, we additionally include CoxKL for a linear Cox student. CoxKL is reported only in this setting because it is defined on Cox partial-likelihood objects and is not a drop-in baseline for DeepNCC (Wang et al., 2026). Component-wise KL gives the strongest stratified discrimination for both DeepSurv and DeepNCC, and achieves the best stratified predictive deviance for Deep-NCC, supporting the claim that aligning teacher and student on locally normalized component distributions is especially useful when the student objective itself is defined on matched comparability sets.

# 6. Discussion

This work studies knowledge transfer in survival and case–control settings where learning is intrinsically locally normalized. In matched designs, Cox-type objectives, and nested case–control sampling, inference is identified through within-set relative-risk contrasts rather than globally normalized outcome probabilities. We therefore formulate

distillation on the induced component distributions and align teacher and student via a component-wise KL divergence, which incorporates teacher predictive information without requiring shared parameterizations, calibrated global probabilities, or access to the external individual-level data used to train the teacher.

Empirically, component-wise distillation is most beneficial in regimes that motivate locally normalized objectives in the first place, including rare events, limited effective sample sizes, and heterogeneous populations. In simulation and in the OPTN case study, the method improves likelihood-based fit and discrimination while remaining robust to changes in teacher score scaling, consistent with transferring relative-risk structure within comparability sets. These results also suggest a practical interpretation of risk-set subsampling in deep survival training as operating on event-centered components that naturally support stochastic optimization and component-level teacher guidance.

The framework has clear boundary conditions. Distillation requires that the teacher can be evaluated on individuals appearing in each comparability set, and the computational cost depends on the structure of the component outcome space, which becomes more complex under ties or multi-event components. Moreover, the distillation weight governs a bias–variance tradeoff. The local expansion in Appendix A.2 formalizes this tradeoff: teacher-aligned component information can reduce estimation variance, but teacher–target mismatch introduces bias, and large weights can amplify negative transfer when the teacher is out of domain. In practice, we tune $\eta$ by five-fold cross-validation on the student cohort, which provides a practical mechanism for downweighting unreliable teacher guidance, but it does not remove the need for target-cohort validation. Future work includes adaptive or component-dependent weighting based on teacher reliability or shift diagnostics, and extensions to more complex censoring and sampling mechanisms.

## Acknowledgements

This work was supported in part by Health Resources and Services Administration contract 234-2005-370011C. The content is the responsibility of the authors alone and does not necessarily reflect the views or policies of the Department of Health and Human Services, nor does mention of trade names, commercial products, or organizations imply endorsement by the U.S. Government. This work was partially supported by the National Institutes of Health grant DK129539.

## Impact Statement

This paper presents work whose goal is to advance the field of survival analysis and machine learning by enabling principled knowledge transfer in censored survival and case–control settings where learning objectives are intrinsically locally normalized (e.g., matched designs, Cox-type risk-set conditioning, and nested case–control sampling). By formulating distillation at the level of design-induced likelihood components, the proposed approach makes it possible to incorporate external predictive information under heterogeneity without requiring shared model parameterizations, calibrated global probabilities, or access to external individual-level data. This can improve risk prediction and decision support in high-stakes applications with limited internal samples and rare events, while promoting reproducible and privacy-conscious reuse of information across studies and institutions.

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

## A. Theorems

### A.1. Derivation of Proposition 4.1

For each component $k$, the student component likelihood is

$$f_k(\mathbf{y}^{(k)}; \boldsymbol{\beta}) = \frac{\exp\left\{\sum_{i \in \mathcal{I}^{(k)}} y_i^{(k)} s_{\boldsymbol{\beta}}(\mathbf{X}_i)\right\}}{Z_k(\boldsymbol{\beta})},$$

and therefore

$$\log f_k(\mathbf{y}^{(k)}; \boldsymbol{\beta}) = \sum_{i \in \mathcal{I}^{(k)}} y_i^{(k)} s_{\boldsymbol{\beta}}(\mathbf{X}_i) - \log Z_k(\boldsymbol{\beta}). \tag{10}$$

The component-wise KL term can be written as

$$\mathrm{KL}\left(\tilde{f}_k \,\|\, f_k(\cdot; \boldsymbol{\beta})\right) = \sum_{\mathbf{y} \in \mathcal{Y}^{(k)}} \tilde{f}_k(\mathbf{y}) \log \tilde{f}_k(\mathbf{y}) - \sum_{\mathbf{y} \in \mathcal{Y}^{(k)}} \tilde{f}_k(\mathbf{y}) \log f_k(\mathbf{y}; \boldsymbol{\beta})$$

$$= C_k - \mathbb{E}_{\tilde{f}_k}\left[\log f_k(\mathbf{Y}^{(k)}; \boldsymbol{\beta})\right], \tag{11}$$

where

$$C_k = \sum_{\mathbf{y} \in \mathcal{Y}^{(k)}} \tilde{f}_k(\mathbf{y}) \log \tilde{f}_k(\mathbf{y})$$

does not depend on $\boldsymbol{\beta}$. Using (10),

$$\mathbb{E}_{\tilde{f}_k}\left[\log f_k(\mathbf{Y}^{(k)}; \boldsymbol{\beta})\right] = \mathbb{E}_{\tilde{f}_k}\left[\sum_{i \in \mathcal{I}^{(k)}} Y_i^{(k)} s_{\boldsymbol{\beta}}(\mathbf{X}_i) - \log Z_k(\boldsymbol{\beta})\right]$$

$$= \sum_{i \in \mathcal{I}^{(k)}} \mathbb{E}_{\tilde{f}_k}\left[Y_i^{(k)}\right] s_{\boldsymbol{\beta}}(\mathbf{X}_i) - \log Z_k(\boldsymbol{\beta})$$

$$= \sum_{i \in \mathcal{I}^{(k)}} \tilde{p}_{ik} s_{\boldsymbol{\beta}}(\mathbf{X}_i) - \log Z_k(\boldsymbol{\beta}). \tag{12}$$

Therefore, dropping constants independent of $\boldsymbol{\beta}$,

$$-\eta \sum_{k=1}^K \mathrm{KL}\left(\tilde{f}_k \,\|\, f_k(\cdot; \boldsymbol{\beta})\right) \propto \sum_{k=1}^K \left[\eta \sum_{i \in \mathcal{I}^{(k)}} \tilde{p}_{ik} s_{\boldsymbol{\beta}}(\mathbf{X}_i) - \eta \log Z_k(\boldsymbol{\beta})\right]. \tag{13}$$

On the other hand, the empirical composite log-likelihood is

$$\ell(\boldsymbol{\beta}) = \sum_{k=1}^K \left[\sum_{i \in \mathcal{I}^{(k)}} y_i^{(k)} s_{\boldsymbol{\beta}}(\mathbf{X}_i) - \log Z_k(\boldsymbol{\beta})\right]. \tag{14}$$

Combining (13) and (14) gives

$$\ell_\eta(\boldsymbol{\beta}) \propto \sum_{k=1}^K \left[\sum_{i \in \mathcal{I}^{(k)}} \left(y_i^{(k)} + \eta \tilde{p}_{ik}\right) s_{\boldsymbol{\beta}}(\mathbf{X}_i) - (1 + \eta) \log Z_k(\boldsymbol{\beta})\right],$$

which is Eq. (8).

Finally, multiplying the objective by the positive constant $(1 + \eta)^{-1}$ does not change its maximizer. Hence Eq. (8) is equivalent to maximizing the original component likelihood with labels

$$\tilde{y}_i^{(k)} = \frac{1}{1 + \eta} y_i^{(k)} + \frac{\eta}{1 + \eta} \tilde{p}_{ik}.$$

This proves Proposition 4.1.

## A.2. Local Excess-Risk Expansion

This section gives a local asymptotic expansion that clarifies the bias–variance role of the component-wise KL penalty. The result is not intended as a universal improvement guarantee; rather, it formalizes a sufficient local regime in which a small positive distillation weight can reduce the expected internal composite risk when the teacher-induced component information is sufficiently aligned with the target population.

For the $n$-sample, let $K_n$ denote the number of composite-likelihood components. For component $k = 1, \ldots, K_n$, let $\mathcal{I}^{(k)}$ be the comparability set and let

$$\mathbf{Y}^{(k)} = \left(Y_i^{(k)}\right)_{i \in \mathcal{I}^{(k)}} \in \mathcal{Y}^{(k)}$$

be the observed label vector. The student component model is

$$f_k(\mathbf{y}^{(k)}; \boldsymbol{\beta}) = \frac{\exp\left\{\sum_{i \in \mathcal{I}^{(k)}} y_i^{(k)} s_{\boldsymbol{\beta}}(\mathbf{X}_i)\right\}}{Z_k(\boldsymbol{\beta})},$$

where

$$Z_k(\boldsymbol{\beta}) = \sum_{\mathbf{u} \in \mathcal{Y}^{(k)}} \exp\left\{\sum_{i \in \mathcal{I}^{(k)}} u_i s_{\boldsymbol{\beta}}(\mathbf{X}_i)\right\}.$$

Let $\tilde{f}_k$ be the teacher-induced component distribution on the same restricted outcome space, and define the teacher-implied component marginal

$$\tilde{p}_{ik} = \mathbb{E}_{\tilde{f}_k}\left[Y_i^{(k)}\right], \qquad i \in \mathcal{I}^{(k)}.$$

For the student model, define the model-implied component marginal

$$p_{ik}(\boldsymbol{\beta}) = \mathbb{E}_{f_k(\cdot; \boldsymbol{\beta})}\left[Y_i^{(k)}\right].$$

Define the normalized internal composite risk

$$R_n(\boldsymbol{\beta}) = -\frac{1}{n}\sum_{k=1}^{K_n} \log f_k(\mathbf{Y}^{(k)}; \boldsymbol{\beta}) = -\frac{1}{n}\sum_{k=1}^{K_n}\left[\sum_{i \in \mathcal{I}^{(k)}} Y_i^{(k)} s_{\boldsymbol{\beta}}(\mathbf{X}_i) - \log Z_k(\boldsymbol{\beta})\right].$$

Up to an additive constant independent of $\boldsymbol{\beta}$, the normalized component-wise KL penalty can be written as

$$Q_n(\boldsymbol{\beta}) = \frac{1}{n}\sum_{k=1}^{K_n}\left[\log Z_k(\boldsymbol{\beta}) - \sum_{i \in \mathcal{I}^{(k)}} \tilde{p}_{ik} s_{\boldsymbol{\beta}}(\mathbf{X}_i)\right].$$

The empirical criterion minimized by the distilled estimator is

$$M_{n,\eta}(\boldsymbol{\beta}) = R_n(\boldsymbol{\beta}) + \eta Q_n(\boldsymbol{\beta}).$$

Let $R$ and $Q$ denote the corresponding population criteria.

The empirical gradients have the forms

$$\nabla R_n(\boldsymbol{\beta}) = \frac{1}{n}\sum_{k=1}^{K_n}\sum_{i \in \mathcal{I}^{(k)}} \left\{p_{ik}(\boldsymbol{\beta}) - Y_i^{(k)}\right\} \nabla s_{\boldsymbol{\beta}}(\mathbf{X}_i),$$

and

$$\nabla Q_n(\boldsymbol{\beta}) = \frac{1}{n}\sum_{k=1}^{K_n}\sum_{i \in \mathcal{I}^{(k)}} \left\{p_{ik}(\boldsymbol{\beta}) - \tilde{p}_{ik}\right\} \nabla s_{\boldsymbol{\beta}}(\mathbf{X}_i).$$

All gradients and Hessians below are understood on the identifiable tangent space at the target parameter $\boldsymbol{\beta}^\star$.

**Assumption A.1** (Local regularity for component-wise distillation). There exists an open neighborhood $\mathcal{U}$ of $\boldsymbol{\beta}^\star$ such that the following conditions hold.

(CL1) **Smoothness and well-defined component distributions.** For every component $k \leq K_n$ and every $i \in \mathcal{I}^{(k)}$, the map $\boldsymbol{\beta} \mapsto s_\beta(\mathbf{X}_i)$ is three times continuously differentiable on $\mathcal{U}$. The teacher component distributions $\tilde{f}_k$ are well defined, so the marginals $\tilde{p}_{ik}$ exist. The induced criteria $R_n, Q_n, R$, and $Q$ are twice continuously differentiable on $\mathcal{U}$.

(CL2) **Local identification of the internal target.** The population internal composite risk $R$ satisfies
$$\nabla R(\boldsymbol{\beta}^\star) = 0, \qquad H := \nabla^2 R(\boldsymbol{\beta}^\star) \succ 0.$$

(CL3) **Stable curvature under small distillation weights.** The population distillation criterion $Q$ satisfies
$$\mathbf{b} := \nabla Q(\boldsymbol{\beta}^\star), \qquad G := \nabla^2 Q(\boldsymbol{\beta}^\star).$$
Moreover, there exist $\bar{\eta} > 0$ and $c_H > 0$ such that
$$\lambda_{\min}(H + \eta G) \geq c_H, \qquad \forall \eta \in [0, \bar{\eta}].$$

(CL4) **Local score expansions.** There exist random vectors $\boldsymbol{\xi}_n, \mathbf{u}_n, \mathbf{v}_n$ such that
$$\nabla R_n(\boldsymbol{\beta}^\star) = n^{-1/2}\boldsymbol{\xi}_n + \mathbf{u}_n, \qquad \mathbb{E}(\boldsymbol{\xi}_n) = 0, \qquad \mathbb{E}(\boldsymbol{\xi}_n\boldsymbol{\xi}_n^\mathsf{T}) \to \Sigma, \qquad \mathbb{E}\|\mathbf{u}_n\|^2 = o(n^{-1}),$$
and
$$\nabla Q_n(\boldsymbol{\beta}^\star) = \mathbf{b} + \mathbf{v}_n, \qquad \mathbb{E}\|\mathbf{v}_n\|^2 = o(1).$$

(CL5) **Local empirical solution control.** Let $\eta_n \to 0$ be any deterministic sequence with $0 \leq \eta_n \leq \bar{\eta}$ for all large $n$. There exists a sequence of local empirical solutions $\widehat{\boldsymbol{\beta}}_{\eta_n}$ of $M_{n,\eta_n}$ such that, with
$$\boldsymbol{\delta}_n := \widehat{\boldsymbol{\beta}}_{\eta_n} - \boldsymbol{\beta}^\star, \qquad a_n := n^{-1} + \eta_n^2,$$
we have $\widehat{\boldsymbol{\beta}}_{\eta_n} \in \mathcal{U}$ with probability tending to one, and
$$\mathbf{g}_n := \nabla M_{n,\eta_n}(\widehat{\boldsymbol{\beta}}_{\eta_n}), \qquad \mathbb{E}\|\mathbf{g}_n\|^2 = o(a_n), \qquad \mathbb{E}\|\boldsymbol{\delta}_n\|^4 = O(a_n^2).$$

(CL6) **Hessian stability along the local path.** Define the averaged empirical Hessians
$$\bar{H}_{n,R} := \int_0^1 \nabla^2 R_n(\boldsymbol{\beta}^\star + t\boldsymbol{\delta}_n)\, dt, \qquad \bar{H}_{n,Q} := \int_0^1 \nabla^2 Q_n(\boldsymbol{\beta}^\star + t\boldsymbol{\delta}_n)\, dt.$$
Then
$$\mathbb{E}\|\bar{H}_{n,R} - H\|^4 = o(1), \qquad \mathbb{E}\|\bar{H}_{n,Q} - G\|^4 = o(1).$$

(CL7) **Mean quadratic expansion of the target risk.** The population internal composite risk satisfies
$$\mathbb{E}\left|R(\widehat{\boldsymbol{\beta}}_{\eta_n}) - R(\boldsymbol{\beta}^\star) - \frac{1}{2}\boldsymbol{\delta}_n^\mathsf{T} H \boldsymbol{\delta}_n\right| = o(a_n).$$

(CL8) **Remainder comparability for direct risk differences.** For every deterministic sequence $\eta_n \to 0$ with $\eta_n = O(n^{-1})$, define
$$A_\eta := (H + \eta G)^{-1},$$
and
$$\rho_n(\eta_n) := \mathbb{E}\left[R(\widehat{\boldsymbol{\beta}}_{\eta_n}) - R(\boldsymbol{\beta}^\star)\right] - \frac{1}{2n}\operatorname{tr}(H A_{\eta_n} \Sigma A_{\eta_n}) - \frac{\eta_n^2}{2}\mathbf{b}^\mathsf{T} A_{\eta_n} H A_{\eta_n} \mathbf{b}.$$
Then
$$\rho_n(\eta_n) - \rho_n(0) = o(n^{-2}),$$
where
$$\rho_n(0) := \mathbb{E}\left[R(\widehat{\boldsymbol{\beta}}_0) - R(\boldsymbol{\beta}^\star)\right] - \frac{1}{2n}\operatorname{tr}(H^{-1}\Sigma),$$
and $\widehat{\boldsymbol{\beta}}_0$ denotes the local empirical solution corresponding to $\eta_n \equiv 0$.

*Remark* A.2. Assumption A.1 places component-wise composite-likelihood distillation in a regular local $M$-estimation regime on the identifiable tangent space at $\beta^\star$. The design-specific objects $\mathcal{I}^{(k)}$, $\mathcal{Y}^{(k)}$, and $Z_k(\beta)$ enter only through the empirical criteria $R_n$ and $Q_n$. Once these criteria satisfy the local score, Hessian, and quadratic-expansion conditions, the expansion applies regardless of whether a component corresponds to a matched stratum, a full Cox risk set, or an NCC sampled risk set. Condition **(CL8)** is used only for the direct comparison between the distilled and internal-only estimators on the $\eta_n = O(n^{-1})$ scale.

**Theorem A.3** (Local excess-risk expansion for component-wise distillation). *Assume Conditions **(CL1)**–**(CL7)** of Assumption A.1. For $\eta \in [0, \bar{\eta}]$, let*

$$A_\eta := (H + \eta G)^{-1}.$$

*Let $\eta_n \to 0$ be any deterministic sequence with $0 \le \eta_n \le \bar{\eta}$ for all large $n$, and define*

$$\delta_n := \widehat{\beta}_{\eta_n} - \beta^\star, \qquad a_n := n^{-1} + \eta_n^2.$$

*Then there exists a remainder vector $\mathbf{s}_n$ such that*

$$\delta_n = -A_{\eta_n} \left\{ n^{-1/2} \xi_n + \eta_n \mathbf{b} \right\} + \mathbf{s}_n, \qquad \mathbb{E}\|\mathbf{s}_n\|^2 = o(a_n). \tag{15}$$

*Consequently,*

$$\mathbb{E}\left[ R(\widehat{\beta}_{\eta_n}) - R(\beta^\star) \right] = \frac{1}{2n} \operatorname{tr}(H A_{\eta_n} \Sigma A_{\eta_n}) + \frac{\eta_n^2}{2} \mathbf{b}^\mathsf{T} A_{\eta_n} H A_{\eta_n} \mathbf{b} + o(a_n). \tag{16}$$

*Moreover, as $\eta \downarrow 0$,*

$$A_\eta = H^{-1} - \eta H^{-1} G H^{-1} + O(\eta^2), \tag{17}$$

*and therefore*

$$\mathbb{E}\left[ R(\widehat{\beta}_{\eta_n}) - R(\beta^\star) \right] = \frac{1}{2n} \operatorname{tr}(H^{-1}\Sigma) - \frac{\eta_n}{n} \Gamma + \frac{\eta_n^2}{2} B + o\left( n^{-1} + \eta_n^2 \right), \tag{18}$$

*where*

$$\Gamma := \operatorname{tr}\left( H^{-1} G H^{-1} \Sigma \right), \qquad B := \mathbf{b}^\mathsf{T} H^{-1} \mathbf{b}.$$

*If, in addition, Condition **(CL8)** holds and $\eta_n = O(n^{-1})$, then*

$$\mathbb{E}\left[ R(\widehat{\beta}_{\eta_n}) - R(\beta^\star) \right] - \mathbb{E}\left[ R(\widehat{\beta}_0) - R(\beta^\star) \right] = -\frac{\eta_n}{n} \Gamma + \frac{\eta_n^2}{2} B + o(n^{-2}). \tag{19}$$

*Hence, if $\Gamma > 0$ and $B > 0$, then for every constant $c \in (0, 2\Gamma/B)$ there exists $n_0 < \infty$ such that, with $\eta_n = c/n$,*

$$\mathbb{E}\left[ R(\widehat{\beta}_{\eta_n}) - R(\beta^\star) \right] < \mathbb{E}\left[ R(\widehat{\beta}_0) - R(\beta^\star) \right], \qquad \forall n \ge n_0. \tag{20}$$

*Remark* A.4. The expansion in Theorem A.3 shows that the local effect of component-wise KL distillation on excess internal composite risk is governed by two competing terms. The term $-(\eta_n/n)\Gamma$ is the first-order change in the variance component; when $\Gamma > 0$, it corresponds to variance reduction from teacher-aligned component information. The term $(\eta_n^2/2)B$ is the distillation-induced bias caused by teacher–target discrepancy at $\beta^\star$. Thus the theorem formalizes the bias–variance tradeoff controlled by $\eta_n$. The strict improvement statement is local and applies in a shrinking-weight regime; it should not be interpreted as a guarantee that any positive distillation weight improves performance under arbitrary teacher mismatch.

*Proof.* By the integral form of Taylor's theorem applied to the vector-valued gradient,

$$\mathbf{g}_n = \nabla R_n(\beta^\star) + \bar{H}_{n,R}\delta_n + \eta_n \left\{ \nabla Q_n(\beta^\star) + \bar{H}_{n,Q}\delta_n \right\}.$$

Using Condition **(CL4)**, this becomes

$$\mathbf{g}_n = n^{-1/2} \xi_n + \mathbf{u}_n + \bar{H}_{n,R}\delta_n + \eta_n \left\{ \mathbf{b} + \mathbf{v}_n + \bar{H}_{n,Q}\delta_n \right\}.$$

Rearranging terms gives

$$(H + \eta_n G)\delta_n = -\left\{ n^{-1/2} \xi_n + \eta_n \mathbf{b} \right\} + \mathbf{e}_n,$$

where
$$\mathbf{e}_n := \mathbf{g}_n - \mathbf{u}_n - \eta_n \mathbf{v}_n - (\bar{H}_{n,R} - H)\boldsymbol{\delta}_n - \eta_n(\bar{H}_{n,Q} - G)\boldsymbol{\delta}_n.$$

We first show that $\mathbb{E}\|\mathbf{e}_n\|^2 = o(a_n)$. By Conditions **(CL4)–(CL5)**,
$$\mathbb{E}\|\mathbf{g}_n - \mathbf{u}_n - \eta_n \mathbf{v}_n\|^2 \leq 3\mathbb{E}\|\mathbf{g}_n\|^2 + 3\mathbb{E}\|\mathbf{u}_n\|^2 + 3\eta_n^2 \mathbb{E}\|\mathbf{v}_n\|^2 = o(a_n).$$

By Conditions **(CL5)–(CL6)** and Cauchy–Schwarz,
$$\mathbb{E}\|(\bar{H}_{n,R} - H)\boldsymbol{\delta}_n\|^2 \leq \left(\mathbb{E}\|\bar{H}_{n,R} - H\|^4\right)^{1/2} \left(\mathbb{E}\|\boldsymbol{\delta}_n\|^4\right)^{1/2} = o(1) \cdot O(a_n) = o(a_n),$$

and
$$\eta_n^2 \mathbb{E}\|(\bar{H}_{n,Q} - G)\boldsymbol{\delta}_n\|^2 \leq \eta_n^2 \left(\mathbb{E}\|\bar{H}_{n,Q} - G\|^4\right)^{1/2} \left(\mathbb{E}\|\boldsymbol{\delta}_n\|^4\right)^{1/2} = o(\eta_n^2 a_n) = o(a_n).$$

Hence $\mathbb{E}\|\mathbf{e}_n\|^2 = o(a_n)$.

Left-multiplying by $A_{\eta_n} = (H + \eta_n G)^{-1}$ and using Condition **(CL3)**, which implies $\|A_{\eta_n}\| \leq c_H^{-1}$ for all large $n$, gives
$$\boldsymbol{\delta}_n = -A_{\eta_n}\left\{n^{-1/2}\boldsymbol{\xi}_n + \eta_n \mathbf{b}\right\} + \mathbf{s}_n, \qquad \mathbf{s}_n := A_{\eta_n}\mathbf{e}_n.$$

Moreover,
$$\mathbb{E}\|\mathbf{s}_n\|^2 \leq c_H^{-2}\mathbb{E}\|\mathbf{e}_n\|^2 = o(a_n).$$

This proves (15).

Next, Condition **(CL7)** gives
$$\mathbb{E}\left[R(\widehat{\boldsymbol{\beta}}_{\eta_n}) - R(\boldsymbol{\beta}^\star)\right] = \frac{1}{2}\mathbb{E}(\boldsymbol{\delta}_n^\mathsf{T} H\boldsymbol{\delta}_n) + o(a_n).$$

Let
$$\mathbf{z}_n := -A_{\eta_n}\left\{n^{-1/2}\boldsymbol{\xi}_n + \eta_n \mathbf{b}\right\}, \qquad \boldsymbol{\delta}_n = \mathbf{z}_n + \mathbf{s}_n.$$

Then
$$\mathbb{E}(\boldsymbol{\delta}_n^\mathsf{T} H\boldsymbol{\delta}_n) = \mathbb{E}(\mathbf{z}_n^\mathsf{T} H\mathbf{z}_n) + 2\mathbb{E}(\mathbf{z}_n^\mathsf{T} H\mathbf{s}_n) + \mathbb{E}(\mathbf{s}_n^\mathsf{T} H\mathbf{s}_n).$$

Because $\|A_{\eta_n}\| \leq c_H^{-1}$ and
$$\mathbb{E}\|\boldsymbol{\xi}_n\|^2 = \mathrm{tr}\{\mathbb{E}(\boldsymbol{\xi}_n\boldsymbol{\xi}_n^\mathsf{T})\} = O(1),$$

we have $\mathbb{E}\|\mathbf{z}_n\|^2 = O(a_n)$. Thus, by Cauchy–Schwarz,
$$\left|\mathbb{E}(\mathbf{z}_n^\mathsf{T} H\mathbf{s}_n)\right| \leq \|H\| \left(\mathbb{E}\|\mathbf{z}_n\|^2\right)^{1/2} \left(\mathbb{E}\|\mathbf{s}_n\|^2\right)^{1/2} = o(a_n),$$

and
$$\mathbb{E}(\mathbf{s}_n^\mathsf{T} H\mathbf{s}_n) \leq \|H\|\mathbb{E}\|\mathbf{s}_n\|^2 = o(a_n).$$

Therefore
$$\mathbb{E}(\boldsymbol{\delta}_n^\mathsf{T} H\boldsymbol{\delta}_n) = \mathbb{E}(\mathbf{z}_n^\mathsf{T} H\mathbf{z}_n) + o(a_n).$$

Expanding the leading term,
$$\mathbb{E}(\mathbf{z}_n^\mathsf{T} H\mathbf{z}_n) = \mathbb{E}\left[\{n^{-1/2}\boldsymbol{\xi}_n + \eta_n \mathbf{b}\}^\mathsf{T} A_{\eta_n} H A_{\eta_n}\{n^{-1/2}\boldsymbol{\xi}_n + \eta_n \mathbf{b}\}\right]$$
$$= \frac{1}{n}\mathrm{tr}\left(A_{\eta_n} H A_{\eta_n}\mathbb{E}(\boldsymbol{\xi}_n\boldsymbol{\xi}_n^\mathsf{T})\right) + \eta_n^2 \mathbf{b}^\mathsf{T} A_{\eta_n} H A_{\eta_n}\mathbf{b},$$

where the cross term vanishes because $\mathbb{E}(\boldsymbol{\xi}_n) = 0$. Since $\mathbb{E}(\boldsymbol{\xi}_n\boldsymbol{\xi}_n^\mathsf{T}) \to \Sigma$ and $\|A_{\eta_n}\| = O(1)$,
$$\frac{1}{n}\mathrm{tr}\left(A_{\eta_n} H A_{\eta_n}\mathbb{E}(\boldsymbol{\xi}_n\boldsymbol{\xi}_n^\mathsf{T})\right) = \frac{1}{n}\mathrm{tr}(A_{\eta_n} H A_{\eta_n}\Sigma) + o(n^{-1}).$$

By cyclicity of the trace,
$$\mathrm{tr}(A_{\eta_n} H A_{\eta_n}\Sigma) = \mathrm{tr}(H A_{\eta_n}\Sigma A_{\eta_n}).$$

Combining the preceding displays proves (16).

To derive (17), write

$$A_\eta = H^{-1/2} \left( I + \eta H^{-1/2} G H^{-1/2} \right)^{-1} H^{-1/2}.$$

For sufficiently small $\eta$, the inverse admits the Neumann expansion

$$\left( I + \eta H^{-1/2} G H^{-1/2} \right)^{-1} = I - \eta H^{-1/2} G H^{-1/2} + O(\eta^2),$$

which implies (17). Consequently,

$$A_\eta H A_\eta = H^{-1} - 2\eta H^{-1} G H^{-1} + O(\eta^2).$$

Therefore,

$$\frac{1}{2n} \operatorname{tr}(H A_\eta \Sigma A_\eta) = \frac{1}{2n} \operatorname{tr}(H^{-1}\Sigma) - \frac{\eta}{n} \operatorname{tr}(H^{-1} G H^{-1}\Sigma) + O(\eta^2/n),$$

and

$$\frac{\eta^2}{2} \mathbf{b}^\mathsf{T} A_\eta H A_\eta \mathbf{b} = \frac{\eta^2}{2} \mathbf{b}^\mathsf{T} H^{-1} \mathbf{b} + O(\eta^3).$$

Substituting these expansions into (16) yields (18), because $\eta_n \to 0$ implies

$$O(\eta_n^2/n + \eta_n^3) = o(n^{-1} + \eta_n^2).$$

Finally, suppose that $\eta_n = O(n^{-1})$ and Condition **(CL8)** holds. By the definition of $\rho_n(\eta_n)$ and Condition **(CL8)**,

$$\rho_n(\eta_n) - \rho_n(0) = o(n^{-2}).$$

Since $A_0 = H^{-1}$, we obtain

$$\mathbb{E}\left[ R(\widehat{\boldsymbol{\beta}}_{\eta_n}) - R(\boldsymbol{\beta}^\star) \right] - \mathbb{E}\left[ R(\widehat{\boldsymbol{\beta}}_0) - R(\boldsymbol{\beta}^\star) \right]$$
$$= \frac{1}{2n} \left\{ \operatorname{tr}(H A_{\eta_n} \Sigma A_{\eta_n}) - \operatorname{tr}(H^{-1}\Sigma) \right\} + \frac{\eta_n^2}{2} \mathbf{b}^\mathsf{T} A_{\eta_n} H A_{\eta_n} \mathbf{b} + o(n^{-2}).$$

Using the expansions above and $\eta_n = O(n^{-1})$,

$$\frac{1}{2n} \left\{ \operatorname{tr}(H A_{\eta_n} \Sigma A_{\eta_n}) - \operatorname{tr}(H^{-1}\Sigma) \right\} = -\frac{\eta_n}{n} \Gamma + O(n^{-3}),$$

and

$$\frac{\eta_n^2}{2} \mathbf{b}^\mathsf{T} A_{\eta_n} H A_{\eta_n} \mathbf{b} = \frac{\eta_n^2}{2} B + O(n^{-3}).$$

Therefore,

$$\mathbb{E}\left[ R(\widehat{\boldsymbol{\beta}}_{\eta_n}) - R(\boldsymbol{\beta}^\star) \right] - \mathbb{E}\left[ R(\widehat{\boldsymbol{\beta}}_0) - R(\boldsymbol{\beta}^\star) \right] = -\frac{\eta_n}{n} \Gamma + \frac{\eta_n^2}{2} B + o(n^{-2}),$$

which proves (19).

If $\eta_n = c/n$, then

$$\mathbb{E}\left[ R(\widehat{\boldsymbol{\beta}}_{\eta_n}) - R(\boldsymbol{\beta}^\star) \right] - \mathbb{E}\left[ R(\widehat{\boldsymbol{\beta}}_0) - R(\boldsymbol{\beta}^\star) \right] = \frac{1}{n^2} \left( -c\Gamma + \frac{c^2}{2} B \right) + o(n^{-2}).$$

If $0 < c < 2\Gamma/B$, the leading coefficient is strictly negative, so there exists $n_0 < \infty$ such that (20) holds for all $n \geq n_0$. $\quad\square$

## B. Evaluation Metrics

Evaluating survival predictions needs to account for censoring. We employed some commonly used metrics to assess model performance.

## B.1. Concordance Index

The concordance index ($C$-index) (Harrell et al., 1982) measures how well a model ranks individuals by risk. Let $\hat{r}(\mathbf{X}_i)$ denote a predicted risk score, where larger values indicate higher risk (shorter survival). A pair $(i, j)$ is comparable if $T_i < T_j$ and subject $i$ experiences an event at $T_i$ (i.e., $\Delta_i = 1$). The $C$-index is the probability that, among comparable pairs, the subject who fails earlier has higher predicted risk:

$$
\begin{aligned}
C &= P\Big(\hat{r}(\mathbf{X}_i) > \hat{r}(\mathbf{X}_j) \,\Big|\, T_i < T_j, \, \Delta_i = 1\Big) \\
&\approx \frac{\sum_{i \neq j} \mathbb{1}[T_i < T_j] \, \mathbb{1}[\hat{r}(\mathbf{X}_i) > \hat{r}(\mathbf{X}_j)] \, \Delta_i}{\sum_{i \neq j} \mathbb{1}[T_i < T_j] \, \Delta_i}.
\end{aligned}
\tag{21}
$$

When ties occur in $\hat{r}(\mathbf{X})$, they can be handled by either excluding tied pairs or assigning partial credit; in our experiments we adopt a consistent tie-handling rule across all methods.

## B.2. Stratified Concordance Index

In matched or stratified designs (e.g., sampled risk sets or matched NCC strata), meaningful comparisons are restricted to units within the same comparability set. We therefore also report a stratified $C$-index that evaluates concordance using only within-stratum comparable pairs.

Let $G_i \in \{1, \ldots, S\}$ denote the stratum (comparability-set) membership for subject $i$. The stratified concordance index is defined as

$$
\begin{aligned}
C_{\mathrm{str}} &= P\Big(\hat{r}(\mathbf{X}_i) > \hat{r}(\mathbf{X}_j) \,\Big|\, T_i < T_j, \, \Delta_i = 1, \, G_i = G_j\Big) \\
&\approx \frac{\sum_{i \neq j} \mathbb{1}[G_i = G_j] \, \mathbb{1}[T_i < T_j] \, \mathbb{1}[\hat{r}(\mathbf{X}_i) > \hat{r}(\mathbf{X}_j)] \, \Delta_i}{\sum_{i \neq j} \mathbb{1}[G_i = G_j] \, \mathbb{1}[T_i < T_j] \, \Delta_i}.
\end{aligned}
\tag{22}
$$

Equivalently, if $N_s$ and $D_s$ denote the numerator and denominator computed using only pairs with $G_i = G_j = s$, then $C_{\mathrm{str}} = \sum_{s=1}^{S} N_s / \sum_{s=1}^{S} D_s$, i.e., a pair-count-weighted aggregation of within-stratum concordance. When no stratification is present (or $G_i$ is constant), $C_{\mathrm{str}}$ reduces to $C$.

## B.3. Time-Dependent AUC

In addition to the $C$-index, we also evaluated model discrimination using the time-dependent area under the ROC curve (AUC). The time-dependent AUC quantifies, at a given time point $t$, how well a model distinguishes individuals who experience an event before or at $t$ (cases) from those who remain event-free beyond $t$ (controls) (Uno et al., 2007; Hung & Chiang, 2010; Lambert & Chevret, 2016). The cumulative dynamic AUC at time $t$ is defined as

$$
\widehat{\mathrm{AUC}}(t) = \frac{\sum_{i=1}^{n} \sum_{j=1}^{n} \mathbb{1}(T_i \leq t, \Delta_i = 1) \mathbb{1}(T_j > t) \widehat{\omega}_i \mathbb{1}\{\hat{r}(\mathbf{X}_j) \leq \hat{r}(\mathbf{X}_i)\}}{\left[\sum_{i=1}^{n} \mathbb{1}(T_i \leq t, \Delta_i = 1) \widehat{\omega}_i\right] \left[\sum_{j=1}^{n} \mathbb{1}(T_j > t)\right]}.
\tag{23}
$$

where $\hat{r}(\mathbf{X}_i)$ is the model's risk score and $\widehat{\omega}_i$ are inverse probability of censoring weights (IPCW) estimated from the training data. A higher value of $\widehat{\mathrm{AUC}}(t)$ indicates better time-specific discrimination.

The MeanAUC aggregates the time-dependent AUC over the full time interval:

$$
\mathrm{MeanAUC} = \frac{1}{\widehat{F}(t_{\max}) - \widehat{F}(t_{\min})} \int_{t_{\min}}^{t_{\max}} \widehat{\mathrm{AUC}}(t) \, d\widehat{F}(t), \quad \widehat{F}(t) = 1 - \widehat{S}(t),
\tag{24}
$$

where $\widehat{S}(t)$ is the Kaplan–Meier estimator of the survival function. This is approximated via numerical integration and implemented in `sksurv`.

## B.4. Integrated Brier Score

The Integrated Brier Score (IBS) evaluates the global calibration and accuracy of time-dependent survival predictions under right censoring. Following the IPCW (inverse probability of censoring weighted) formulation from Graf et al. (1999);

Kvamme et al. (2019); Gerds & Schumacher (2006), the Brier score at time $t$ is defined as

$$\text{BS}(t) = \frac{1}{n} \sum_{i=1}^{n} W_i(t) \left( \hat{S}(t \mid \mathbf{X}_i) - \mathbb{1}(T_i > t) \right)^2,$$

(25)

where $\hat{S}(t \mid \mathbf{X}_i)$ denotes the predicted survival probability and $W_i(t)$ represents the IPCW weight constructed from the Kaplan–Meier estimator of the censoring distribution. This weighting corrects for the loss of observed outcome status due to right censoring under standard independent-censoring assumptions and yields an unbiased estimate of prediction error.

The IBS aggregates the Brier score over the full time interval:

$$\text{IBS} = \frac{1}{t_{\max} - t_{\min}} \int_{t_{\min}}^{t_{\max}} \text{BS}(t) \, dt,$$

(26)

which is approximated via numerical integration. A lower IBS indicates better calibration and discrimination of survival predictions across time.

### B.5. Predictive Deviance

Predictive deviance is defined as the design-specific negative log-likelihood, or negative composite log-likelihood, of the fitted model on the test data; lower values indicate better likelihood-based fit (Burnham & Anderson, 2002; Tutz et al., 2016). For donor-matched evaluation, stratified predictive deviance is computed by aggregating the negative log-likelihood over the donor-matched comparability sets used for evaluation.

## C. Additional Experiment Settings

### C.1. Additional Implementation Details

**Implementation.** All methods are implemented in Python 3.9 with PyTorch 1.12.1. Experiments follow a teacher–student workflow. We first train a teacher model, which may be fit under a different study design than the student. The trained teacher is then evaluated on the student cohort to generate per-individual risk scores used for distillation. Finally, the student is trained on the target cohort by minimizing the design-specific composite objective augmented with our component-wise distillation term, which performs alignment within the same comparability sets used by the student likelihood. Across methods, we keep data splits, tuning budgets, and early-stopping rules fixed to ensure comparability. Unless stated otherwise, teacher and student use the same backbone network, an MLP with residual connections. Training is capped at 128 epochs and uses early stopping with patience 5, where one epoch corresponds to a full pass over the training set.

**Tuning Parameters.** Architecture and optimization settings are selected using five-fold cross-validation with predictive deviance on the held-out fold as the model-selection criterion. Automated search is performed with Optuna (Akiba et al., 2019) using the Tree-structured Parzen Estimator (TPE) sampler (Bergstra et al., 2011; 2013; Watanabe, 2023). Each optimization run uses a fixed budget of 20 trials. For each trial, the model is trained on four folds and evaluated on the held-out fold; we retain the configuration achieving the lowest mean validation predictive deviance across the five folds.

The search domain includes the number of hidden layers in $\{1, 2, 3, 4\}$, hidden width in $\{32, 64, 128\}$, dropout rate in $[0.1, 0.2]$, batch size in $\{32, 64, 128\}$, and learning rate in $[5 \times 10^{-4}, 10^{-3}]$; batch normalization is used in all runs. For distilled models, we additionally tune the distillation weight $\eta \in [0, 20]$.

Teacher and student hyperparameters are tuned separately on their respective non-test data. After teacher hyperparameters are selected, the teacher is refit on the full non-test teacher data and then evaluated on the student cohort to generate per-individual risk scores used for distillation. For the student, we compared joint tuning of network hyperparameters and $\eta$ against a two-stage procedure that tunes network hyperparameters first and then tunes $\eta$ with the network fixed. Since the two procedures gave similar performance in preliminary experiments, we use the two-stage procedure for lower computational cost and clearer separation of distillation strength from model capacity and optimization.

Because the objectives are defined on comparability sets, mini-batches are formed by sampling sets and including all individuals within each selected set. We report a target batch size in terms of individuals per gradient step, but the realized batch size depends on the sampled number of sets and their design-specific set sizes. When set sizes vary, we choose the number of sampled sets so that the realized batch size is as close as possible to the target.

## C.2. Additional Simulation Settings

### C.2.1. SIMULATED NESTED CASE–CONTROL DATASETS

We simulate covariates $\boldsymbol{X} = (X_1, \ldots, X_{12})^\top$ with independent components $X_j \sim \mathcal{N}(0, 1)$. We form three summary features $Z_1 = \sum_{j=1}^4 X_j$, $Z_2 = \sum_{j=5}^8 X_j$, and $Z_3 = \sum_{j=9}^{12} X_j$. Event times are generated from an exponential model with a nonlinear risk function,

$$T^* \sim Exp\left((\beta_1 Z_1)^2 + (\beta_2 Z_2)^2 + (\beta_3 Z_3)^2\right), \tag{27}$$

with $(\beta_1, \beta_2, \beta_3) = (8, 10, 12)$. Independent censoring times are drawn as $C \sim \mathcal{U}(0, 100)$, and we observe $T = \min\{T^*, C\}$ with event indicator $\Delta = \mathbb{1}(T^* \leq C)$. Under this configuration, the marginal event rate is approximately 9.3%.

We generate three cohorts from the same data-generating process: a teacher cohort ($n = 10{,}000$), a student cohort ($n = 500$), and an independent test cohort ($n = 5{,}000$). To construct NCC training data, we apply risk-set sampling to the teacher and student cohorts. For each observed failure at time $t_i$ from case $i$, we sample $n_c = 16$ controls without replacement from the corresponding risk set $R(t_i) \setminus \{i\}$, yielding a comparability set with exactly one event and $n_c$ non-events. The resulting collection of sampled sets defines the NCC composite-likelihood components used to fit models under an NCC objective. The test cohort is not subjected to NCC sampling; all evaluation is performed on the full test cohort to reflect population-level predictive performance.

**Evaluation Setup.** We compare three approaches: (i) an internal-only baseline trained on the student NCC data, (ii) a teacher trained on the teacher NCC data, and (iii) a distilled student trained on the student NCC data using component-wise distillation with guidance from the teacher. Unless otherwise stated, teacher and student share the same architecture and use the same NCC construction, so differences primarily reflect the effect of component-wise distillation rather than changes in optimization or likelihood components. Each experiment is repeated over 20 independent random seeds.

### C.2.2. DATA-GENERATING PROCESS WITH COVARIATE SHIFT

To study teacher–student covariate shift, we generate the student cohort as in Appendix C.2.1 and generate the teacher cohorts from a shifted covariate distribution.

We retain the same covariate partition as in Appendix C.2.1, with $\boldsymbol{X} = (X_1, \ldots, X_{12})^\top$ grouped into three blocks $\boldsymbol{X}_{1:4}$, $\boldsymbol{X}_{5:8}$, and $\boldsymbol{X}_{9:12}$. For each block $g \in \{1, 2, 3\}$, we draw an independent binary mask $\boldsymbol{m}_g \in \{0, 1\}^4$ with i.i.d. entries $m_{g\ell} \sim \text{Bernoulli}(p)$ and sample

$$\boldsymbol{X}_g \sim \mathcal{N}(\boldsymbol{m}_g s, \; \boldsymbol{I}_4),$$

where $s$ controls the shift magnitude and $p$ controls the expected fraction of shifted coordinates within each block.

Let $Z_1 = \sum_{j=1}^4 X_j$, $Z_2 = \sum_{j=5}^8 X_j$, and $Z_3 = \sum_{j=9}^{12} X_j$. We generate a latent failure time from an exponential model with a nonlinear predictor,

$$T^* \sim Exp\left((\beta_1 Z_1)^2 + (\beta_2 Z_2)^2 + \sum_{j=9}^{12} (\beta_3 X_j)^2\right).$$

Relative to Appendix C.2.1, this replaces the block-level squared term for $\boldsymbol{X}_{9:12}$ with a separable sum-of-squares term. We then sample censoring times independently as $C \sim \mathcal{U}(0, c_{\max})$ and set $T = \min\{T^*, C\}$ with $\Delta = \mathbb{1}(T^* \leq C)$. Unless otherwise noted, we use $(\beta_1, \beta_2, \beta_3) = (8, 10, 12)$ and $c_{\max} = 100$.

We generate two shifted teacher datasets (each with $n = 10{,}000$): (i) a moderate shift with $(s, p) = (-0.3, 0.3)$ and (ii) a severe shift with $(s, p) = (1.0, 0.6)$. These regimes induce different incidence and censoring profiles; empirically, the event rates are 5.66% (moderate shift) and 3.08% (severe shift).

In both regimes, the teacher is trained on the shifted population, whereas the student is trained on the original cohort using the same NCC risk-set sampling and fitting procedure as in Appendix C.2.1. This setup isolates the effect of teacher–student covariate shift on distillation.

# D. Additional Experiment Analysis

## D.1. Simulation

### D.1.1. EFFECT OF DISTILLATION WEIGHT $\eta$

The distillation weight $\eta$ controls the relative strength of component-wise distillation compared with the student likelihood and is therefore a key tuning parameter. When the teacher is informative, larger $\eta$ can leverage stronger guidance and improve generalization in small student cohorts. When the teacher is weak or misspecified, overly large $\eta$ can over-weight unreliable guidance and degrade performance. We therefore sweep $\eta$ to assess sensitivity and to characterize how the preferred distillation strength varies with teacher quality induced by covariate restriction.

Figure 6 summarizes the results. As $\eta$ increases, the student transitions from the internal-only baseline ($\eta = 0$) toward a more teacher-driven one, with performance typically improving and then saturating. The best-performing $\eta$ depends on teacher quality: stronger teachers support larger $\eta$ and yield larger gains, whereas weaker teachers favor smaller $\eta$, with limited but generally non-negative improvements over internal-only training.

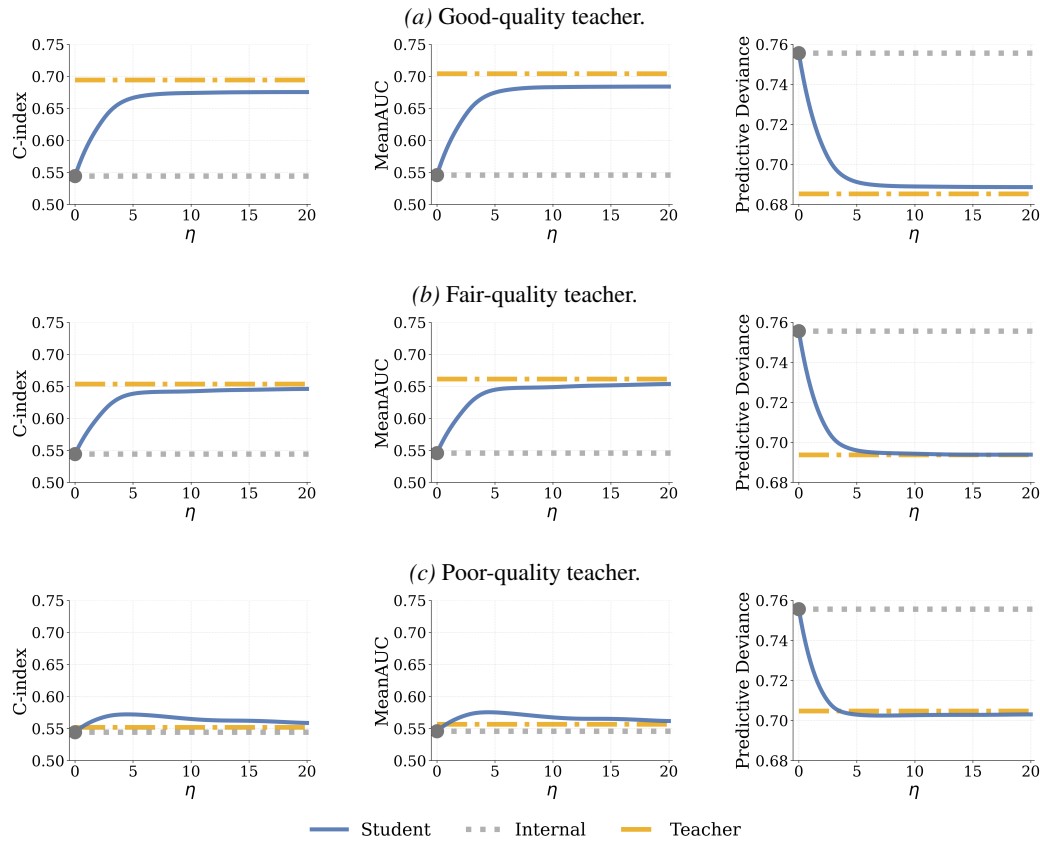

*Figure 6.* Test-set C-index, mean time-dependent AUC, and predictive deviance across varying values of the weighting parameter $\eta$ under different teacher covariate quality settings. Results were averaged over 20 independent simulation replicates.

