# OpenReview forum: "Component-Wise Composite Likelihood Distillation for Censored Time-to-Event Data"
_ICML.cc/2026/Conference — ICML 2026 regular_

### Official Review · Reviewer_HQ42 · 2026-03-05

**Soundness:** 2
**Presentation:** 2
**Significance:** 3
**Originality:** 2
**Overall Recommendation:** 4
**Confidence:** 3

**Summary:**

This paper addresses the challenge of applying knowledge distillation to censored time-to-event data, specifically in settings governed by locally normalized objectives such as Cox partial likelihoods and nested case-control (NCC) designs. The authors identify a fundamental mismatch between standard distillation methods—which typically rely on globally normalized probabilities—and survival models where inference is driven by within-set relative risks and nuisance parameters are conditioned out.

To bridge this gap, the manuscript proposes a Component-Wise Composite Likelihood Distillation framework. This approach decomposes the learning objective into design-induced "comparability sets" (e.g., risk sets or matched strata) and aligns the teacher and student models by minimizing the Kullback–Leibler divergence between their induced distributions within each specific component. A key advantage of this formulation is that it transfers predictive relative-risk structure without requiring the teacher to be globally calibrated or to share the same parameterization as the student. The authors validate the method through simulations and a real-world case study using the OPTN kidney transplant registry, demonstrating that the proposed distillation improves discrimination and calibration in data-limited, rare-event, and heterogeneous settings.

**Compliance With Llm Reviewing Policy:**

Affirmed.

**Final Justification:**

The authors have adequately addressed my main concerns in the rebuttal. I have increased my score accordingly

**Key Questions For Authors:**

1. Regarding the Definition of the Restricted Outcome Space $Y_k(1)$ in Section 4.3:

    In your formulation of the Cox Partial Likelihood (Section 4.3), you define the restricted outcome space $Y_k(1)$ as the set of all possible binary vectors where exactly one failure occurs within the risk set $R(\tau_k)$. However, in the actual observed data for a given risk set at time $\tau_k$, only a single specific individual is observed to fail.

    Question: How do you mathematically reconcile the definition of $Y_k(1)$—which implies a distribution over potential failures—with the fact that the observation is a fixed singleton? Is your framework treating the observed failure as a realization from this hypothetical distribution, and if so, does this interpretation hold when the risk set size $|R(\tau_k)|$ varies dynamically?

    Impact: Clarifying this is critical to determining if your "component-wise" probability space is theoretically consistent with the standard partial likelihood interpretation, or if it introduces a logical contradiction between the sample space and the observed realization.

2. Derivation of Equation (8) and "Additive Constants":

    In Section 4.2, you state that Equation (8) is derived by "expanding the component-wise KL terms and dropping additive constants that do not depend on $\beta$."

    Question: Could you explicitly provide the intermediate steps showing how the term $(1+\eta)\log Z_k(\beta)$ emerges and why the teacher's normalization term $\tilde{Z}_k$ is entirely absorbed into the constant? Specifically, does the "constant" assumption hold even if the teacher's scoring function $\tilde{s}(X)$ has a different scale or shift compared to the student's initialization?

    Impact: I found the jump from Eq. (5) to Eq. (8) abrupt. A clear derivation is necessary to verify the mathematical soundness of the loss function and to ensure that the gradient flow correctly balances the empirical evidence and the distillation penalty.

3. Missing Comparison with CoxKL:

    You explicitly mention CoxKL (Wang et al., 2023) in your Related Work as a method that "regularizes a Cox model via a KL divergence defined on Cox partial-likelihood objects." This appears to be the most direct competitor to your proposed method.

    Question: Why was CoxKL excluded from the empirical baselines in Section 5? Can you provide a performance comparison (e.g., C-index) between your Component-Wise Distillation and CoxKL?

    Impact: Without this comparison, it is impossible to assess whether your method offers a genuine improvement over the existing state-of-the-art. If CoxKL performs similarly or better, the justification for your more complex "composite likelihood" framework weakens significantly.

**Limitations:**

The authors have not adequately discussed the limitations and potential negative societal impact of their work. To improve the manuscript, I suggest adding a dedicated discussion covering the following points:

1. Theoretical Limitations: The paper lacks an analysis of the convergence properties of the proposed component-wise optimization. It is unclear under what conditions the student model is guaranteed to converge to the teacher's relative risk ordering, or how the method behaves when the teacher's risk estimates are noisy or biased.

2. Scalability and High-Dimensionality: The empirical validation relies heavily on low-dimensional simulations ($d=12$). The authors should acknowledge that the performance in high-dimensional settings (common in modern genomics or EHR analysis) remains unverified.

3. Societal Impact & Fairness: Given the biomedical focus (e.g., organ transplantation), the authors should discuss the risk of transferring algorithmic bias from the Teacher to the Student. If the Teacher model was trained on biased historical data, the distillation process might amplify these biases in the Student model, potentially affecting equitable healthcare delivery.

**Strengths And Weaknesses:**

1. Strengths

1.1 Significance & Originality: The paper addresses a genuine and non-trivial problem in survival analysis: how to perform knowledge distillation when global probability distributions are not identifiable due to censoring and nuisance parameters. The core idea of aligning models at the "component" (risk set) level rather than the global level is conceptually appealing and intuitively aligns with the structure of partial likelihood inference.

1.2. Real-World Application: The inclusion of the OPTN kidney transplant case study demonstrates an attempt to ground the theoretical framework in a high-stakes, practical domain where data heterogeneity is a common challenge.

2. Weaknesses

2.1 Soundness (Mathematical Logic & Derivations): The mathematical foundation of the proposed framework appears fragile and, at times, logically confused.

      2.1.1. Derivation Gaps: The transition from the component-wise KL divergence (Eq. 5) to the specific loss formulation in Equation (8) is abrupt and lacks rigorous justification. The authors claim that additive constants "do not depend on $\beta$" and can be dropped, but the derivation glosses over the complex dependencies between the normalization term $Z_k(\beta)$ and the scoring function $s_\beta$. The proportionality claim in Eq. (8) feels hand-waved rather than mathematically proven.

     2.1.2. Nuisance Parameter Cancellation: The central premise relies on the cancellation of the shared term $b_k$. While this is a known property of the Cox partial likelihood, the paper attempts to generalize this to a broader "composite likelihood" framework without reasonble definition of the restricted space $\mathcal{Y}_{k}(d_{k})$, see my question in Key Question Section.

     2.1.3. Evaluation Rigor: The empirical evaluation is statistically weak. The results lack confidence intervals, p-values, or error bars across multiple random seeds. Furthermore, the absence of strong baselines (such as CoxKL or other domain adaptation methods) makes it impossible to verify if the mathematical complexity actually yields performance gains over existing techniques.

2.2. Presentation (Clarity & Polish): The presentation of the manuscript is significantly below the standard expected for a top-tier conference.

       2.2.1.Obscure Writing: The narrative is dense and often confusing. The authors frequently introduce terms (e.g., "comparability sets," "restricted outcome space") without intuitive explanations, forcing the reader to struggle through the logic. The connection between the "general" composite likelihood theory and the specific implementation in NCC is muddy.

       2.2.2. Unverifiable Citations: The paper relies on self-citations to "Authors (2026a)" and "Authors (2026b)" to support key claims. Since these are concurrent, anonymous submissions, the reader cannot verify the validity of the foundational work this paper builds upon, creating a circular and opaque argumentative structure.

       2.2.3.Significance: While the problem is important, the impact of this specific work is severely limited by the confusion in the methodological explanation. If the mathematical derivation is not rigorous and the presentation is riddled with errors, the community is unlikely to adopt or build upon this framework. The lack of clarity regarding why this method works mathematically (beyond intuition) limits its theoretical contribution.

---

> ### Author Rebuttal · Authors · 2026-03-31
>
> We thank the reviewer for the careful reading. We agree that the main issues are the definition of the component-level probability object, the derivation from Eq.(5) to Eq.(8), the need for a direct comparison with CoxKL, and the need to broaden the empirical scope.
>
> **Restricted outcome space in Cox partial likelihood.** For the Cox case, $\mathcal Y^{(k)}(1)$ is a conditional sample space, not a claim that multiple failures are simultaneously observed. At event time $\tau_k$, the component is the risk set $R(\tau_k)$ and $$\mathcal{Y}^{(k)}(1)=\lbrace \mathbf{y}\in \lbrace0,1\rbrace^{\lvert R(\tau_k)\rvert}: \sum_{i\in R(\tau_k)} y_i=1 \rbrace.$$ This is exactly the standard conditional interpretation of Cox partial likelihood: given $R(\tau_k)$ and one failure at $\tau_k$, the model assigns the conditional probability of which member fails. The observed singleton is one realization from this component-specific space. The fact that $|R(\tau_k)|$ varies with $k$ simply means that the comparability set and restricted outcome space are component-dependent.
>
> **Scope of nuisance-term cancellation.** Our claim is not that arbitrary nuisance terms cancel. The construction applies when the within-component score has the form $$u_k(i;\boldsymbol{\beta}) = b_k + s_{\boldsymbol{\beta}}(\mathbf X_i),$$
> with $b_k$ shared by all admissible outcomes in component $k$. Then $b_k$ appears in both numerator and denominator and cancels under local normalization. This is the structure used in Eq.(1) and in the CLR/Cox/matched-NCC special cases, and the KL term is defined on that same induced conditional distribution.
>
> **Explicit derivation of Eq.(8).** We agree that this step should have been written out. For each component,
> $$\log f_k(\mathbf y^{(k)};\boldsymbol{\beta}) = \sum_{i \in \mathcal I^{(k)}} y_i^{(k)} s_{\boldsymbol{\beta}}(\mathbf X_i) - \log Z_k(\boldsymbol{\beta}),$$ and $$\mathrm{KL}\left(\tilde f_k \|\| f_k(\cdot;\boldsymbol{\beta})\right) = \sum_{\mathbf y \in \mathcal Y^{(k)}} \tilde f_k(\mathbf y)\log \tilde f_k(\mathbf y) - \sum_{\mathbf y \in \mathcal Y^{(k)}} \tilde f_k(\mathbf y)\log f_k(\mathbf y;\boldsymbol{\beta}).$$ Substituting this into the component KL gives $$\mathrm{KL}\left(\tilde f_k \|\| f_k(\cdot;\boldsymbol{\beta})\right) = C_k - \sum_{i \in \mathcal I^{(k)}} \tilde p_{ik} s_{\boldsymbol{\beta}}(\mathbf X_i) + \log Z_k(\boldsymbol{\beta}),$$ where $\tilde p_{ik}=\mathbb E_{\tilde f_k}[Y_i^{(k)}]$ and $C_k$ collects all terms independent of $\beta$, including teacher-only normalization. Summing over $k$ yields Eq.(8) up to an additive constant. This remains true even if the teacher score has a different location or scale, because once $\tilde f_k$ is formed, all teacher-only normalization terms are fixed with respect to $\beta$.
>
> **Comparison with CoxKL.** We agree that CoxKL is the most relevant baseline in Cox-compatible settings. We therefore added CoxKL to the donor-matched recipient-only analysis and report the updated comparison in **Table 3** (https://anonymous.4open.science/r/supp0A6F/Table%203.md). CoxKL improves over the internal-only Cox baseline (stratified C-index: $0.5467$ vs. $0.5382$; stratified deviance: $0.2128$ vs. $0.2156$), but remains below our component-wise KL in stratified C-index ($0.5875$). More importantly, CoxKL is tied to Cox partial-likelihood objects, whereas our contribution is a distillation principle for design-induced locally normalized objectives, including NCC settings.
>
> **Additional real data analysis.**  We also broadened the empirical scope by adding a multiple myeloma transfer analysis under TT3$\rightarrow$TT2, a distinct high-dimensional regime from OPTN. **Table 2** (https://anonymous.4open.science/r/supp0A6F/Table%202.md) shows the same qualitative pattern: the distilled student improves over the internal-only student and also exceeds the teacher on the target cohort (C-index: $0.6803$ vs. $0.5842$ internal-only and $0.6659$ teacher; predictive deviance: $1.8867$ vs. $2.4999$ and $1.8995$).
>
> **Theoretical clarification.** We also now add an asymptotic result showing, informally, that under local regularity conditions and $\eta_n=O(n^{-1})$, component-wise distillation strictly improves the expected target composite risk whenever the first-order teacher-alignment gain dominates the second-order bias term. This is not a universal guarantee under arbitrary mismatch, but it formalizes the intended regime of the paper. We provide a fuller statement in our response to Reviewer vwTA and will include the theorem and proof in the appendix.
>
> **Summary.** In revision, we will make the conditional interpretation of $\mathcal Y^{(k)}$, the algebra from Eq.(5) to Eq.(8), and the scope of the nuisance-cancellation claim explicit in the main text. Together with the added CoxKL comparison, the TT3$\rightarrow$TT2 analysis, and the new asymptotic result, these changes address the reviewer's concerns about mathematical clarity, empirical comparability, and scope.

---

> > ### Author Rebuttal · Reviewer_HQ42 · 2026-03-31
> >
> > The authors have adequately addressed my main concerns in the rebuttal. I have increased my score accordingly

---

> > > ### Author Response · Authors · 2026-04-05
> > >
> > > We sincerely appreciate the reviewer’s thoughtful reconsideration and are glad that the additional clarifications were helpful.

---

### Official Review · Reviewer_4G6Q · 2026-03-10

**Soundness:** 2
**Presentation:** 3
**Significance:** 2
**Originality:** 2
**Overall Recommendation:** 3
**Confidence:** 4

**Summary:**

This paper proposes a Component-Wise Composite Likelihood Distillation method for censored time-to-event data. The key idea is to use the KL divergence between component-wise distributions to distill knowledge from a teacher model without requiring the full probability distribution. The method is evaluated through simulation studies under multiple mismatch scenarios and a real-world organ transplant case study.

**Compliance With Llm Reviewing Policy:**

Affirmed.

**Ethical Review Concerns:**

The paper cites concurrent submissions in ICML 2026 and provide filenames.

**Ethical Review Flag:**

Flag this paper for an ethics review.

**Ethics Expertise Needed:**

["Privacy and Security (e.g., personally identifiable information)"]

**Final Justification:**

I appreciate the clarification on scope. The conceptual connection to deep survival subsampling and ranking/retrieval is reasonable, but no experiment validates distillation outside the classical Cox/NCC regime. Therefore, I maintain my score but would not object to acceptance.

**Key Questions For Authors:**

1. The framework relies entirely on the proportional hazards assumption. Many modern deep survival models use discrete-time formulations with globally normalized outcome probabilities, where standard distillation applies directly. Can you discuss when your locally normalized approach offers advantages over simply distilling in a discrete-time framework?

2. The distillation penalty scales as $O(K)$, the same order as the composite log-likelihood $O(n)$. This means $\eta$ cannot be treated as a conventional hyperparameter independent of sample size. Can you provide analysis or guidance on how $\eta$ should scale with n and K?

3. The real-data evaluation is limited to a single dataset (OPTN). Could you validate on additional datasets with different event rates, censoring patterns, or study designs to demonstrate generalizability?

**Limitations:**

yes

**Strengths And Weaknesses:**

Strengths:
1. This paper addresses an important and common setting in biomedical survival analysis, where globally normalized outcome probabilities are not identifiable but locally normalized probabilities are, and proposes a distillation-based approach to leverage such locally normalized information.
2. The presentation is clean and well-structured.

Weaknesses:
1. The framework relies entirely on the proportional hazards assumption, with no discussion of extensions to non-PH settings, limiting its practical applicability. For example, many modern deep survival models adopt discrete-time formulations where globally normalized outcome probabilities are identifiable, making standard distillation applicable.
2. Usually the penalty is $O(1)$ and dominated by the $O(n)$ likelihood, the distillation penalty scales as $O(K)$, the same order as the composite log-likelihood. Thus, $\eta$ cannot be treated as a conventional hyperparameter. However the paper provides no analysis of its relationship with $n$ and $K$.
3. Lack of experiments on real-world datasets. The paper is validated on only one real dataset (OPTN), which limits the generalizability of the empirical results.

---

> ### Author Rebuttal · Authors · 2026-03-31
>
> We thank the reviewer for the thoughtful comments. We agree that the key issues are: (i) when a locally normalized distillation framework is needed relative to standard discrete-time distillation, (ii) how the distillation weight $\eta$ should be interpreted when both the likelihood and the KL term are aggregated over components, and (iii) whether the empirical evidence extends beyond the OPTN case study.
>
> First, our framework is not tied to a single proportional-hazards model. The organizing principle is broader: learning under design-induced comparability sets through locally normalized conditional, partial, or composite objectives. This includes matched and stratified conditional models, Cox-type risk-set objectives, and sampled risk-set designs such as NCC. The common feature is that, because of conditioning, matching, sampling, or nuisance cancellation, the student is identified through within-set comparisons rather than through direct fitting of one globally normalized outcome model for each individual.
>
> We agree with the reviewer that when the student is a globally normalized discrete-time survival model fit directly to per-individual interval outcome probabilities, then standard response-based distillation can be appropriate. Our contribution is aimed at the complementary regime. In many survival and case--control settings, discretizing time does not by itself remove the study-design constraints that determine what the student objective can actually learn. Under conditional / partial / composite likelihood training, the relevant learned object is generally not a global outcome distribution, but the collection of component-wise conditional distributions induced on the restricted outcome spaces of the comparability sets. This is why our transfer target is defined at the component level. We will revise the paper to make this boundary clearer: standard discrete-time KD is suitable when the student directly optimizes a globally normalized outcome model, whereas our framework is designed for settings in which the student objective remains locally normalized even after time discretization.
>
> Second, we agree that the discussion of $\eta$ should be sharper. The reviewer is right that, in the raw summed objective, both the composite log-likelihood and the KL term scale with the number of components, so $\eta$ should not be interpreted independently of the chosen component construction and normalization. The clean interpretation is therefore through the component-averaged objective, or equivalently the mini-batch objective used in optimization, where both terms are averaged over sampled components. Under this normalization, $\eta$ does not need to grow mechanically with $K$; rather, it controls the local tradeoff between empirical labels and teacher guidance. This is also explicit in Eq.(8), where the objective is equivalent to using soft labels, so $\eta$ directly determines the interpolation between the observed component label and the teacher-implied component target. In common regimes, $K$ grows with the effective sample size because components are induced by events or matched/sampled sets, which further reinforces that the relevant interpretation of $\eta$ is under per-component normalization rather than as a sample-size-free constant. We will revise the paper to state this normalization explicitly.
>
>
>
> Third, we agree that broader real-data evidence would strengthen the paper. To address this directly, we have added a complementary real-data transfer analysis on a **multiple myeloma dataset**, using the larger TT3 cohort as the external source and the TT2 cohort as the internal target. This provides a substantially different biomedical setting from OPTN, with 20,502 gene-level covariates and a different event/censoring regime. The additional results in **Table 2** (https://anonymous.4open.science/r/supp0A6F/Table%202.md) show that the  qualitative conclusion is the same as in OPTN: the distilled student substantially improves over the internal-only student and also slightly outperforms the teacher on the target cohort. Specifically, the C-index improves from $0.5842$ to $0.6803$, compared with $0.6659$ for the teacher, and predictive deviance improves from $2.4999$ to $1.8867$, compared with $1.8995$ for the teacher. We will include this additional analysis in the supplement and summarize it briefly in the main text to demonstrate that the method is not specific to the transplant-registry application.
>
>
> Overall, we hope this clarification better positions the contribution of the paper. The method is intended for student objectives learned through component-wise conditional likelihoods induced by study design, sampling, or conditioning. In such settings, the relevant transfer object is not an externally imposed global predictive distribution, but the component-induced conditional distribution that matches what the student objective can actually learn.

---

> > ### Author Rebuttal · Reviewer_4G6Q · 2026-04-02
> >
> > Thank you for the responses. The clarification on $\eta$ normalization is convincing, and the additional myeloma dataset strengthens the empirical support. However, the fundamental scope concern remains. The method appears to be primarily designed for a locally normalized regime (e.g., matched designs, NCC, and stratified models), which is more commonly associated with classical biostatistics than with mainstream machine learning.

---

> > > ### Author Response · Authors · 2026-04-05
> > >
> > > We thank the reviewer for this important scope comment. We agree that matched designs, NCC sampling, and stratified Cox-type objectives are classical motivating examples. Our point, however, is not that these specific study designs are themselves mainstream machine-learning tools, but that they instantiate a broader machine-learning regime: training and transfer under locally normalized, set-conditioned objectives.
> > >
> > >
> > > This regime also arises in scalable deep survival training through risk-set subsampling and batch-restricted normalization, and it is structurally related to ranking/retrieval/recommendation settings where learning is defined on candidate sets rather than a single globally normalized output space. The contribution of our paper is therefore at the level of a transfer principle for locally normalized objectives, with censored survival and case-control designs serving as the main motivating and empirically validated instances.
> > >
> > > We agree that the current draft can make this positioning more explicit, and we will revise the introduction and related work accordingly. More broadly, we hope this perspective will motivate exploration of component-wise distillation in other set-conditioned learning problems beyond the biomedical settings studied here.

---

### Official Review · Reviewer_vwTA · 2026-03-12

**Soundness:** 3
**Presentation:** 3
**Significance:** 3
**Originality:** 3
**Overall Recommendation:** 4
**Confidence:** 4

**Summary:**

The paper introduces a knowledge distillation framework that leverages external predictive information without requiring the sharing of individual-level data, specifically targeting settings where outcome distributions are only partially specified, such as the censored time-to-event data. The approach utilizes a composite-likelihood with the penalty as the component-wise Kullback-Leibler (KL) divergence to align teacher and student models, since some parameters are not identifiable from within-set comparisons.

**Compliance With Llm Reviewing Policy:**

Affirmed.

**Final Justification:**

The reply resolved my concern

**Key Questions For Authors:**

1. The experiments show that the distilled student model typically improves upon the internal-only student across good, fair, and poor quality teacher models. What is the exact covariate feature set for the fixed student model in these scenarios? Furthermore, if the teacher has fewer covariates than the student (i.e., worse than the student), does the distillation process still provide benefit in practice?

2. In the experimental setup for distilling from the teacher in the presence of covariate shift, is the test cohort drawn from the exact same distribution as the student cohort? Clarification on this would help interpret the robustness of the results.

3. The distillation weight $\eta$ carries a significant burden in balancing the objectives. While automated search handles its selection, the degradation of performance with poor teachers highlights the risk of "negative transfer". Since the paper notes that the estimator is essentially trained on a set of softened labels, should there also be explicit constraints or assumptions regarding distributional shifts for the outcomes (i.e., ensuring the outcomes are not shifted too drastically in the student cohort relative to the teacher cohort)?

**Limitations:**

yes

**Strengths And Weaknesses:**

Strength
1. The paper successfully unifies the view of locally normalized learning over comparability sets, cleanly encompassing objectives from classical survival analysis (e.g., Cox partial likelihood) and case-control analysis (e.g., matched designs).

2. The proposed component-wise KL distillation framework is highly practical, as it remains compatible with flexible deep models trained via stochastic optimization.

3. The initial results from the simulation studies and biomedical case studies demonstrate promising improvements in discrimination and estimation efficiency.

Weakness
1. The paper relies heavily on empirical validation and lacks a **formal theorem** or asymptotic proof demonstrating that the distilled estimator $\beta_{\eta}$ will strictly outperform the internal-only estimator $\beta_{0}$ under the proposed objective.

---

> ### Author Rebuttal · Authors · 2026-03-31
>
> We thank the reviewer for the careful reading and for highlighting the need for a more explicit theoretical justification. We agree that the main issues are to clarify the teacher-quality and covariate-shift experiments, explain how the method handles possible negative transfer, and provide a formal asymptotic result.
>
> First, the teacher-quality simulation already includes the case where the teacher has fewer covariates than the student. The student feature set is fixed at $\{X_1,\ldots,X_{12}\}$ in all scenarios, while teacher quality is varied by restricting the teacher to $\{X_1,\ldots,X_{12}\}$, $\{X_1,\ldots,X_9\}$, or $\{X_1,\ldots,X_6\}$. This is intentional: the teacher is not meant to replace the student's richer covariates, but to provide additional within-set relative-risk information learned from a larger external cohort. This reflects a realistic regime in which the external teacher may be trained on many more individuals but with fewer recorded variables, while the local student has richer covariates but fewer observed events. The simulation shows that such a teacher can still help when it induces informative within-set ordering, although the gain attenuates as the teacher becomes less informative.
>
>
> Second, in the covariate-shift experiment, only the teacher population is shifted. The student training cohort and the evaluation cohort are both generated from the original target-side distribution; only the teacher is trained on a shifted source population. Thus, Figure 3 is designed to isolate teacher--target mismatch, rather than joint out-of-domain shift of both the student and the test cohort. We agree this should be stated more explicitly in the main text.
>
>
> Third, regarding $\eta$ and negative transfer, we agree that this is a real boundary of the method. Our claim is not that tuning $\eta$ eliminates negative transfer under arbitrary teacher quality or source--target shift. Rather, the student is always trained against the target-cohort composite likelihood, while the KL term adds teacher guidance only through component distributions defined on the same comparability sets. Whether this helps depends on how well those teacher-induced component distributions align with the target structure. This is why $\eta$ is selected on the student validation split. Empirically, as the teacher becomes more mismatched, the selected $\eta$ decreases and the gains attenuate, which is consistent with validation-driven downweighting of a less reliable teacher.
>
>
>
> Finally, we agree that the paper should include a formal theorem rather than relying mainly on empirical evidence. In revision, we will add an asymptotic result for the component-wise distillation estimator. The result is not a universal claim of improvement under arbitrary mismatch; instead, it gives a method-specific sufficient condition under which component-wise distillation improves the expected target composite log-loss.
>
> More specifically, let $R(\beta)$ denote the target composite-risk criterion, let $\beta^\star$ be its oracle minimizer, and let $\hat\beta_{\eta_n}$ be the empirical estimator under component-wise distillation with weight $\eta_n$. Under local regularity conditions, score expansion, Hessian control, and an additional local remainder-stability condition on the $\eta_n = O(n^{-1})$ scale, we will show that
>
> $$\mathbb{E}\left[R(\hat\beta_{\eta_n}) - R(\beta^\star)\right]=\frac{1}{2n}\operatorname{tr}(H^{-1}\Sigma)-\frac{\eta_n}{n}\Gamma+\frac{\eta_n^2}{2}B+o\left(n^{-1}+\eta_n^2\right)$$
>
> where $\Gamma$ is the first-order variance-reduction coefficient induced by the teacher-aligned component KL term and $B$ is the corresponding second-order shrinkage-bias coefficient. Moreover, under the additional remainder-stability condition, we obtain the direct comparison
>
> $$\mathbb{E}\left[R(\hat\beta_{\eta_n})-R(\beta^\star)\right]-\mathbb{E}\left[R(\hat\beta_{0})-R(\beta^\star)\right]=-\frac{\eta_n}{n}\Gamma+\frac{\eta_n^2}{2}B+o(n^{-2})$$
>
> Hence, if $\Gamma>0$ and $B>0$, then there exists a nonempty interval
>
> $$0<c<\frac{2\Gamma}{B}$$
>
> such that choosing $\eta_n=c/n$ yields
>
> $$\mathbb{E}\left[R(\hat\beta_{\eta_n})-R(\beta^\star)\right]<\mathbb{E}\left[R(\hat\beta_{0})-R(\beta^\star)\right]$$
>
> for all sufficiently large $n$.
>
> We believe this is the right formal lens for the paper. It does not claim universal improvement, but it does provide an explicit asymptotic sufficient condition for strict improvement and directly formalizes the regime seen in the experiments: when teacher--target alignment is stronger, the admissible gain region is larger; when the teacher is poorer or more shifted, the gain shrinks and the preferred $\eta$ becomes smaller. We will add the formal theorem and proof in the appendix and summarize its main implication in the main text.

---

> > ### Author Rebuttal · Reviewer_vwTA · 2026-04-03
> >
> > Thank you for the reply. The teacher-quality and covariate-shift experiments make sense to me. The claim regarding the composite-risk criterion also seems convincing. I will raise my score to a positive one.

---

> > > ### Author Response · Authors · 2026-04-05
> > >
> > > We thank the reviewer for the careful follow-up and are glad that our clarifications addressed the main concerns.

---

### Official Review · Reviewer_cqsr · 2026-03-13

**Soundness:** 3
**Presentation:** 2
**Significance:** 3
**Originality:** 3
**Overall Recommendation:** 4
**Confidence:** 2

**Summary:**

The paper presents a knowledge distillation framework to address the case where the outcome distributions are partially specified. It uses composite-likelihood Kullbak-Leibler divergence to establish likelihood components. Each likelihood component captures the probability model on a restricted outcome space.

**Compliance With Llm Reviewing Policy:**

Affirmed.

**Final Justification:**

I decided to maintain my score after reading the author's rebuttal.

**Key Questions For Authors:**

1. Can the authors clarify what does it mean by a locally normalized learning objective (in contrast to a globally normalized objective)? As this is a key concept mentioned a lot of times in the introduction. Is it about learning the likelihood of a subpopulation?
2. For Eq (2) (and (4), analogously), is learning $Z_k$ the most critical part rather than learning $d_\beta$? What happens if $Z_k$ is intractable?
3. In Figure 3, the predictive deviance is not changing much between moderate and severe heterogeneous teachers. The paper thus summarizes that the component-wise distillation tolerates covariate shift. Can the author explain the intuition behind it?

**Limitations:**

yes

**Strengths And Weaknesses:**

Soundness: The paper’s method is well-constructed and experiments are sound. The characterizations of CLR, Cox, and NCC look correct to me. The metrics (C-index, MeanAUC, IBS, predictive deviance) used for evaluation are comprehensive.

Presentation: The introduction may need to be more specific. It may be better to present the NCC sampling as a motivation first (as an example for partial observed information) to establish the problem. The method section appears to be clear to me.

Significance: Learning partial information on covariate distribution when covariates are not available for all study periods is an important practical problem. The paper provides a novel method to address this problem and the experimental results look convincing.

Originality: Using KL divergence-based approach to capture local likelihood looks novel in this setting. The novelty is also well justified in the related work.

---

> ### Author Rebuttal · Authors · 2026-03-31
>
> We thank the reviewer for the positive assessment of the paper's soundness, novelty, and practical significance. We also appreciate the suggestion on presentation. We agree that the introduction can motivate the problem more concretely, and in revision we will introduce nested case--control (NCC) earlier as a running example and define ''locally normalized'' explicitly at first use.
>
> By ''locally normalized'', we do not mean learning the likelihood of a subpopulation. The distinction is about where normalization occurs. In a globally normalized model, probabilities are defined on a single global outcome space. In our setting, the study design induces comparability sets $I^{(k)}$ and restricted outcome spaces $\mathcal{Y}^{(k)}(t_k)$, and normalization occurs only within each component. In Cox partial likelihood, this is the conditional probability of the observed failure configuration within a risk set; in matched case--control and NCC settings, it is the conditional distribution on the admissible event patterns within a matched or sampled set. Terms shared by all units in a set, such as baseline hazards, stratum intercepts, or set-level sampling factors, cancel after conditioning. Thus the student objective identifies within-set relative-risk structure rather than globally calibrated absolute risk. This is exactly why our distillation target is defined on the induced component distributions in Eqs.(2)--(6), rather than on a global probability target that is generally not identified by the student objective.
>
> The learned object is the score function $s_\beta(\cdot)$, equivalently the student parameter $\beta$, not $Z_k$ itself. The quantity $Z_k(\beta)$ is the normalizing constant induced by the within-set scores on the restricted outcome space $\mathcal{Y}^{(k)}(t_k)$, and $\tilde Z_k$ plays the analogous role for the teacher-induced component distribution. In other words, we do not separately learn $Z_k$ as a free object. If $Z_k$ were intractable, then the underlying component likelihood would already be difficult to evaluate; this is a boundary condition of the component model, not a separate difficulty introduced by distillation. In the main settings studied here, the component normalization is tractable by design: in Cox without ties it reduces to the usual risk-set sum, in NCC each sampled set is small, and in matched CLR the case count is fixed within a stratum. We agree that ties or larger multi-event components increase this cost, and we will make this scope condition more explicit.
>
> For Figure 3, we agree that the severe-shift setting deserves a more careful interpretation. The intended claim is not that severe shift leaves performance unchanged, but that component-wise distillation remains effective under moderate shift, with gains attenuating as the teacher becomes increasingly out of domain. This attenuation is clearest in C-index and MeanAUC, while the change is visually less pronounced for predictive deviance. To make this explicit, we additionally summarize the two settings in **Table 1** (https://anonymous.4open.science/r/supp0A6F/Table%201.md). The table shows  that the predictive-deviance improvement is smaller under severe shift than under moderate shift ($0.0305$ vs. $0.0390$), even though the visual difference is subtler than for discrimination. The same pattern appears more strongly in C-index and MeanAUC, where the gains drop from $0.0442/0.0469$ under moderate shift to $0.0080/0.0089$ under severe shift. Moreover, the selected distillation strength is smaller under severe shift ($4.20$ vs. $8.70$ on average), indicating that validation automatically chooses weaker transfer when the teacher is more out of domain. Our interpretation is therefore that stronger teacher--target mismatch mainly weakens the transferable within-set ordering signal, which directly affects discrimination, while predictive deviance and IBS are also anchored by the student's own fit to the target-cohort composite objective. We will revise the wording accordingly so that the paper states tolerance to moderate shift, with attenuation rather than invariance under more severe mismatch.
>
> We appreciate the reviewer's positive assessment. We believe the central clarification is that, because the student objective identifies component-level conditional structure rather than a global outcome distribution, our distillation is formulated at that same component level.

---

> > ### Author Rebuttal · Reviewer_cqsr · 2026-04-03
> >
> > I thank the authors for their clarifications and decide to maintain the current positive score.

---

> > > ### Author Response · Authors · 2026-04-05
> > >
> > > We thank the reviewer for the careful follow-up and for updating the assessment.

---

### Decision · Program_Chairs · 2026-04-30

**Decision:**

Accept (regular)

**Comment:**

This is a "classical" borderline paper, for which the reviewers have identified several strengths and weaknesses. On the positive side, most reviewers agree that the paper is technically sound, addresses and interesting problem, contains some novel ideas, and that it is easy to read. On the negative side, several points of criticism have been raised, such as  the lack of a formal theorem demonstrating the benefits of the distilled estimator proposed,  limitations in scope (only proportional hazards assumption), missing real-world experiments and problems in the experimental evaluation (missing confidence intervals etc.). Most of the more technical concerns, however, could be addressed in a rather convincing way during the rebuttal and discussion phase, and I think that for this paper, finally the positive aspects outweigh the problems. Therefore I recommend (weak) acceptance.